# Quantifying the impact of emission outbursts and non-stationary flow on eddy covariance CH₄ flux measurements using wavelet techniques

5  Mathias Göckede[1,*], Fanny Kittler[1], Carsten Schaller[1,2]

[1]Max Planck Institute for Biogeochemistry, Jena, Germany
[2]now: University of Münster, Institute of Landscape Ecology, Climatology Research Group, Münster, Germany

[*]*Correspondence to*: Mathias Göckede (mathias.goeckede@bgc-jena.mpg.de)

**Abstract.** Methane flux measurements by the eddy-covariance technique are subject to large uncertainties, particularly linked to the partly highly intermittent nature of methane emissions. Outbursts of high methane emissions, termed event fluxes, hold the potential to introduce systematic biases into derived methane budgets, since under such conditions the assumption of stationarity of the flow is violated. In this study, we investigate the net impact of this effect by comparing
eddy-covariance fluxes against a wavelet-derived reference that is not negatively influenced by non-stationarity. Our results demonstrate that methane emission events influenced 3-4 % of the flux measurements, and did not lead to systematic biases in methane budgets for the analyzed summer season; however, the presence of events substantially increased uncertainties in short-term flux rates. The wavelet results provided an excellent reference to evaluate the performance of three different gapfilling approaches for eddy-covariance methane fluxes, and we show that none of them could reproduce the range of
observed flux rates. The integrated performance of the gapfilling methods for the longer-term dataset varied between the two eddy-covariance towers involved in this study, and we show that gapfilling remains a large source of uncertainty linked to limited insights into the mechanisms governing the short-term variability in methane emissions. With the capability to broaden our observational methane flux database to a wider range of conditions, including the direct resolution of short term variability at the order of minutes, wavelet-derived fluxes hold the potential to generate new insight into methane exchange
processes with the atmosphere, and therefore also improve our understanding of the underlying processes.

## 1 Introduction

The eddy covariance (EC) technique, a well-established method for the direct quantification of turbulent surface–atmosphere exchange processes (Aubinet et al., 2012), can provide valuable information on current CH₄ flux rates between various types of ecosystems and the atmosphere (e.g. Taylor et al., 2018; Rößger et al., 2019; Tuovinen et al., 2019), including insights
into processes and controls (e.g. Pirk et al., 2016; Kittler et al., 2017b; Neumann et al., 2019) that can be used to improve future projections. However, the data quality of EC measurements depends strongly on the adherence to several theoretical assumptions such as e.g. steady-state conditions and horizontal homogeneity (Foken, 2017), which frequently limits data

availability. In case of methane fluxes, particularly a potential violation of the required steady-state conditions linked to episodic outbursts from wetland sources (Schaller et al., 2019) may lead to low flux data quality, and therefore can substantially increase the gap fraction in quality filtered EC time series.

One potential mechanism for such high methane emission events is so-called ebullition (e.g. Kwon et al., 2017; Peltola et al., 2018; Männistö et al., 2019), i.e. periodic bubble outgassing with a typical length of seconds to minutes. Even though such emissions are part of the natural flux signal, and should therefore be accounted for when accumulating longer-term budgets of methane exchange, in the context of EC data processing and quality assessment these events are likely to be discarded during the quality screening of raw data, or they may be incorrectly handled by the data processing algorithms. In both cases, the natural high flux event would be incorrectly accounted for, potentially introducing systematic biases into methane fluxes and budgets (Baldocchi et al., 2012).

Spatial heterogeneity in the emission patterns of methane surrounding the flux tower (e.g. Rey-Sanchez et al., 2019) may also lead to pronounced variability in the observed $CH_4$ flux time series (Tuovinen et al., 2019). Particularly for wetland ecosystems, ecosystem characteristics such as inundation level or vegetation composition may vary at finest spatial scales (Muster et al., 2012; McEwing et al., 2015), creating microsite variability with strong gradients in methane emissions. Also at landscape (Peltola et al., 2015) to regional scales (Davidson et al., 2016), spatial variability in landscape characteristics may have a strong influence on the captured flux signal. For flux towers situated in such structured areas, an emission spike in the $CH_4$ flux time series can also be created by a temporary shift of the field of view of the sensors from a low flux region into a high flux region and back (e.g. Korrensalo et al., 2018). As outlined for the ebullition fluxes above, depending on the exact nature of the spike the flux signal may be misinterpreted by the eddy-covariance processing software.

Since 'outburst events' in methane fluxes are in many cases flagged as non-stationary conditions, and therefore discarded as low-quality data, the assessment of the net impact of this effect needs to consider what will happen to the resulting gaps in the quality-filtered EC time series. Gaps are a common feature in eddy-covariance time series, resulting e.g. from power failures, instrument malfunctioning, or low data quality linked to the violation of the above-mentioned theoretical assumptions (e.g. Foken et al., 2004). If they can be filled with a reliable, unbiased algorithm, additional gaps would not pose a major problem. For $CO_2$, several of such well-established frameworks are available (e.g. Reichstein et al., 2005; Moffat et al., 2007), allowing to generate continuous time series for the assessment of long-term flux budgets. In contrast, for $CH_4$ fluxes no consensus on a gap-filling method has yet emerged within the EC-community. Several studies succeeded in establishing data-driven links between $CH_4$ fluxes and environmental conditions such as e.g. peat/soil temperature, friction velocity or water table (e.g. Wille et al., 2008; Zona et al., 2009; Jackowicz-Korczynski et al., 2010) using both linear and non-linear functional relationships. Other approaches include gap interpolation (e.g. Rinne et al., 2007; Tagesson et al., 2012), process-based modeling (Forbrich et al., 2011) or artificial neural networks (e.g. Dengel et al., 2013). Even though these different approaches have been shown to perform well in case studies, a solution that has been proven to be uniformly applicable is lacking, therefore large uncertainties are still associated with $CH_4$ gapfilling.

As an alternative to the regular eddy-covariance raw data processing, the flux calculations can also be performed based on the wavelet method by analyzing frequency patterns in the underlying time series of winds and scalars (Collineau and Brunet, 1993b, a). In contrast to the eddy-covariance method, the wavelet method is not restricted by the same set of theoretical assumptions, and in particular no steady-state conditions are required (e.g. Daubechies, 1990). Wavelets have been demonstrated to be a powerful tool for quantifying turbulent fluxes (Mauder et al., 2007; Thomas and Foken, 2007). The ability to calculate turbulent fluxes for periods as short as one minute has been proven very valuable for attributing flux variability to environmental controls, both based on aircraft campaigns (Metzger et al., 2013) and stationary tower measurements within a heterogeneous landscape (Xu et al., 2017). Moreover, wavelet techniques have been applied to improve the frequency correction with the eddy-covariance method (Nordbo and Katul, 2013). A direct comparison between fluxes processed with the wavelet and eddy-covariance method, respectively, found an excellent agreement between both methods for EC data of highest quality (Schaller et al., 2017).

Here, we quantify the net impact of failing to resolve methane outburst events with the EC method, comparing both short-term emission patterns and longer-term flux budgets to a reference flux product derived with wavelet methods. The presented study is closely linked to two recently published manuscripts (Schaller et al., 2017; 2019) that demonstrate that fluxes during such outburst events, with timescales at the order of only a few minutes, can be precisely quantified using the wavelet method, while the coarser temporal resolution of the EC method normally fails to resolve these details while aggregating over 30 minutes. In this follow-up study, we determine systematic offsets between both methods, and the specific role that different types of short-term outburst events play in this context. Since many non-stationary events were leading to data gaps in the EC flux time series, we placed a specific focus on evaluating the performance of different gap-filling algorithms to fill these gaps. Overall, our study aims at evaluating the effect of non-stationary conditions on the long-term methane flux budgets, with a special focus placed on systematic biases introduced by either flux processing approach or chosen gap-filling method.

## 2 Material and Methods

### 2.1 Site description

The Ambolikha research site (Göckede et al., 2017), located on a floodplain of the Kolyma River approximately 18 km south of the town of Chersky, northeast Russia, is underlain by continuous permafrost and characterized as wet tussock tundra dominated by tussock-forming *Carex appendiculata* and *lugens* and *Eriophorum angustifolium* (Corradi et al., 2005; Kwon et al., 2016). Alluvial mineral soils (silty clay) are topped by an organic peat layer (0.15–0.20 m), with some of the organic material also present in deeper layers following cryoturbation (Corradi et al., 2005; Merbold et al., 2009). Averaged for the period 1960 – 2009, the mean annual air temperature was -11°C and the average annual precipitation summed up to 197 mm (Göckede et al., 2017). Vegetation height was ~ 0.7 m during the peak of the growing season, reached around the beginning of August.

Data were collected from two eddy-covariance towers situated about 600 m apart, both elevated ~6 m above sea level. While one measurement system was placed within a drainage ditch system (tower 1, 68.61 °N and 161.34 °E, FLUXNET code RU-Che), therefore capturing fluxes that represent a patch of tundra affected by a lowered water table, the second control measurement system (tower 2, 68.62 °N and 161.35 °E, RU-Ch2) measures natural exchange conditions unaffected by the hydrological disturbance. For this study, a dataset covering the period June 01 to September 18, 2014, was used.

## 2.2 Instrument setup

Both flux towers mentioned above in Section 2.1 were equipped with the same instrumentation, including a sonic anemometer (uSonic-3 Scientific, 5 W heating, METEK GmbH, Elmshorn, DE) at the tower top (at heights of 4.9 m and 5.1 m for drained and control tower, respectively) and a closed-path greenhouse gas analyzer for $CH_4/CO_2/H_2O$ (FGGA, Los Gatos Research Inc., CA, USA). Ambient air was drawn by an external vacuum pump (membrane pump, N940, KNF, 13 L min$^{-1}$ under ambient pressure) from an inlet placed next to the sonic anemometer (vertical sensor separation: 0.30 m) through a heated and insulated sampling line (Eaton Synflex decabon with 6.2 mm inner diameter and a length of 16 m and 13 m for drained and control tower, respectively). The acquisition of high frequency (20 Hz) raw data was handled by the software package EDDYMEAS (Kolle and Rebmann, 2007) on a local computer at the field site.

Ancillary meteorological data were collected at 10 s intervals from both towers and stored as 10-minute averages on a data logger (CR3000, Campbell Scientific, UT, USA). Acquired parameters include e.g. air temperature and humidity, air pressure, precipitation, or soil temperatures. Low-frequency meteorological data underwent a thorough data quality control screening, and subsequently were averaged to 30 minutes (see Kittler et al. (2016) for details).

## 2.3 Raw data processing

We based the raw data processing to obtain fluxes from the collected high frequency data on two different methods:

1. The eddy-covariance raw data processing uses the software package TK3 (Mauder and Foken, 2015). When applied in stand-alone mode, this tool implements all required conversions, corrections, and quality assessment procedures (Foken et al., 2012; Fratini and Mauder, 2014). Details on the TK3 implementation on the Ambolikha datasets are provided by Kittler et al. (2016; 2017a).

2. The second flux processing method (Schaller et al., 2017; 2019) is based on wavelet analysis and uses the sinusoidal and complex valued Morlet wavelet transform for flux quantification. The Morlet wavelet provides an excellent resolution in the frequency domain and can be used to analyze atmospheric turbulence (e.g. Strunin and Hiyama, 2004; Thomas and Foken, 2005). Since this study focused on comparing eddy covariance- and wavelet-derived fluxes, the temporal integration of the wavelet method was chosen to closely match the eddy covariance method (30 min); however, due to the decomposition in time and frequency domain the averaging intervals could not match perfectly, and an averaging interval of 33 min for the wavelet method was used. A detailed description of the wavelet method, the wavelet transform and the corresponding flux data processing can be found in appendix A and in Schaller et al. (2017).

In the context of the presented study, in a first processing step both methods were applied to produce continuous time series of uncorrected half-hour fluxes of methane. In a subsequent processing stage, the results provided by both methods underwent the same flux correction procedure by the TK3 software package, including 2D coordinate rotation of the wind field, cross-wind correction (Liu et al., 2001), and correction for losses in the high-frequency range (Moore, 1986).

The eddy covariance post-processing quality control is commonly based on the analysis of stationary and well-developed turbulence conditions (e.g. Foken et al., 2004; 2012). When applied for wavelet fluxes, the test for stationarity can be dropped, since wavelet flux data quality is not compromised by non-stationary conditions (see above). The development of the turbulence is investigated based on the concept of flux-variance similarity (Wyngaard et al., 1971) via the so-called integral turbulence characteristics (ITC, Foken and Wichura, 1996). A low data quality rating by the ITC can e.g. be caused

by stable atmospheric stratification that suppresses turbulent motions. Data stationarity is tested by comparing signal covariance at different averaging intervals (e.g. 5-minutes vs. 30-minutes, Foken and Wichura, 1996). In this context, effects such as e.g. spikes in the signal, abrupt changes of the signal level, or intermittent turbulence may trigger low flux data quality. We grouped eddy covariance fluxes into different categories (Table 1) based on their stationarity flag (SF) ratings. Fluxes outside the range -10 nmol $m^{-2}$ $s^{-1}$ < $CH_4$ flux < 150 nmol $m^{-2}$ $s^{-1}$ (based on the 2.5 % and 97.5 % quantiles for high

and medium quality $CH_4$ fluxes from tower 2) were sorted out during the post-processing.

Gaps of the eddy covariance time series were filled with three different methods. A linear interpolation (LI) represents the simplest method. The mean of a 10-day moving window (MW) centered to the gap was used for a better representation of the seasonality. These two methods were chosen since they do not require a sophisticated tool, and can thus be easily applied. Finally, a neuronal network approach (NN, Dengel et al., 2013) represents a more sophisticated gapfilling algorithm,

filling gaps based on prevailing environmental conditions.

To assess the agreement between EC and wavelet fluxes, a regression analysis was applied. With flux data of both methods being subject to uncertainties, no independent variable could be identified. Thus, in place of ordinary least-square regression, an orthogonal regression  (OR, linear model II regression) was used with the R-package "lmodel2" (Legendre, 2014) and Pearson's correlation coefficients (r) are given. OR is particularly suited for the comparison of time series that are both

subject to errors of about the same order of magnitude (e.g. Foken, 2017).

**Table 1: Quality flag categories based on the stationarity rating of the eddy covariance flux data. The definition of quality categories follows the scheme proposed by Sabbatini et al. (2018), which is based on stationarity tests developed by Foken et al. (2004; 2012), but uses stricter thresholds to separate categories.**

| Quality | Stationarity flag (SF) | Range of differences [%]* |
|---------|------------------------|---------------------------|
| High | < 3 | 0 – 30 |
| Medium | 3 – 5 | 31 – 100 |
| Low | >5 | > 100 |

*Difference [%] between the covariances calculated over 30 minutes and calculated as a average of six 5-minute covariances (for details please refer to Foken and Wichura, 1996).*

## 2.4 Event characterization

The characterization of high methane emission event types differentiated within the context of this study is based on a wavelet approach using the Mexican Hat wavelet. In contrast to the Morlet wavelet, which we used to precisely quantify flux rates due to its excellent localization in the frequency domain, the Mexican Hat wavelet has a very good localization in the time domain, therefore facilitated an exact localization of single events. Event periods resolved at minute intervals were identified by the median absolute deviation (MAD, e.g. Hoaglin et al., 2000) test followed by an additional manual adjustment. Events were separated into the three categories introduced by Schaller et al. (2019):

1. Peak events: This simple event starts from a baseline flux level, monotonically changes towards a peak or plateau, and subsequently monotonically changes back to the baseline level again.

2. Updown/Downup events: Similar to two connected peak events with opposite sign, after reaching a first peak the fluxes overshoot the baseline level to reach a second peak in the opposite direction before approaching the baseline again. An up-down event indicates a positive peak followed by a negative one, for a down-up event the sequence would be vice versa.

3. Cluster events: Prolonged periods containing numerous high methane emission events were labeled as cluster events. Such periods showed a distinctive pattern of high emissions, compared to the baseline fluxes before and after the event, but did not display the clearly defined peak structures as defined above.

## 3 Results

### 3.1 Data coverage and overall quality flags

Of the 5280 half-hourly flux values that would provide continuous data coverage within the study period June 01 to September 18, 2014, about 3.4 % or 6 % of the eddy fluxes were either missing or discarded as lowest data quality for tower 1 and tower 2, respectively (Table 2). For the wavelet datasets, missing flux values had a slightly higher percentage compared to the EC dataset, since a required 3-hour window of continuous data focusing on the current timestamp

broadened the window of missing fluxes around every gap in the raw data. From the remaining data, a further 11.8 % (tower 1) or 6.6 % (tower 2) were discarded during the EC quality control procedure as low quality, in all cases linked to non-stationary flow conditions. Since the wavelet method does not require stationarity, no additional gaps due to low data quality occurred. For both methods, the subsequent range test (see Section 2.3 for details) filtered out another 2 – 5 % of data that was assigned high to medium quality. Taken together, for each combination of tower and processing method, more than 80 % of the fluxes remained after quality screening. Compared to the wavelets, this percentage is lower by about 7 % for the EC method, linked to the requirement of stationary flow conditions.

Regarding the distribution of gaps over time, no seasonal patterns were found for both towers and both flux processing methods, so each part of the study period received about equal data coverage. With respect to diurnal patterns in gap distribution, EC data display a higher gap fraction during the night, compared to daytime data coverage. This imbalance is most pronounced for tower 1 (see also Appendix B, Figures A1 & A2). No such diurnal patterns in gap distribution were found within the wavelet flux time series, and also no systematic differences between both towers were found for this method. Taken together, wavelet flux data processing provides a better overall data coverage, i.e. less data gaps have to be filled to replace unreliable measurements flagged as low quality. Also, the equal distribution of gaps between day and night supports an improved performance of gap-filling algorithms. Since gapfilled fluxes are associated with higher uncertainties than measured fluxes, this indicates that wavelet data processing holds the potential to produce more robust flux budgets.

**Table 2: Gap fraction within the dataset used for this study, separated by flux processing method and tower position.**

| Gaps [%] | Tower 1 | | Tower 2 | |
|---|---|---|---|---|
| | EC | Wavelet | EC | Wavelet |
| missing data, or lowest quality in raw dataset | 3.43 | 4.13 | 5.95 | 6.93 |
| low quality flag during post-processing | 11.80 | - | 6.59 | - |
| range test flag for remaining medium/high quality data | 3.39 | 5.09 | 2.22 | 2.52 |
| TOTAL SUM | 18.62 | 9.22 | 14.75 | 9.45 |

### 3.2 Flux data quality analysis

### 3.2.1 Comparison of non-gapfilled methane fluxes under different stationarity conditions

In this section, we compare measured methane flux rates between EC and wavelet methods, with the intention to derive the dependence of differences between methods on the stationarity of the underlying flow conditions. This analysis excludes gapfilled results. A comparison between methods focusing on the derivation of long-term flux budgets, which include also the gapfilled values, will be presented in the following Section 3.2.2. For both flux processing methods, higher methane emissions under all quality and stationarity conditions are observed at tower 2 (see also Figure 1), which features a higher fraction of inundated areas in its footprint in comparison to tower 1. At tower 2, for both flux processing methods, flux rates

display a pronounced increase in mid-July, leading to a peak in August and a subsequent decrease at the beginning of September, a pattern that follows the general seasonal trends in soil temperatures. During these times of increased methane emissions, a diurnal cycle with higher flux rates during daytime is observed (see also Figure A1), while at tower 1, for both flux processing methods no seasonal or diurnal cycle was observed.

High stationarity (SF < 3). Under highly stationary flow conditions, we found an excellent agreement between half-hourly flux rates derived with EC and wavelet flux processing, respectively. A direct comparison shows that both methods produce highly correlated $CH_4$-fluxes throughout the spectrum of absolute values, a fact that is confirmed by a orthogonal regression analysis (Wavelet = Intercept + Slope · EC) that produces slopes close to 1, intercepts close to 0 nmol $m^{-2}$ $s^{-1}$, and correlation coefficients of 0.98 for both towers (Table 3). Averaging data under high stationarity for the entire study period yields flux

rates that are only marginally higher for the EC method, compared to the wavelet reference (Table 3, Figure 1).

Medium stationarity (SF 3 – 5). The correlation between half-hourly flux rates derived with both processing methods is reduced under medium stationary flow conditions, compared to the high stationarity. This observation is confirmed by the OR analysis, which produces coefficients that deviate stronger from the ideal targets as shown above for high stationarity (Table 3). Mean flux rates are reduced in comparison to highly stationary conditions, and positive offsets between fluxes

derived by the EC method and the wavelet method, respectively, are higher for both towers (Table 3, Figure 1).

**Table 3: Statistical coefficients of an orthogonal regression analysis (Wavelet = Intercept + Slope · EC) and mean flux rates for the two processing methods separated by stationarity classes (SF<3 as high, SF 3 – 5 as medium, and SF>5 as low stationarity).**

| | Data | N | Intercept [nmol $m^{-2}$ $s^{-1}$] | Slope | r | Mean flux rate [nmol $m^{-2}$ $s^{-1}$] | |
| | | | | | | Wavelet | EC |
|---|---|---|---|---|---|---|---|
| Tower 1 | SF < 3 | 2728 | 0.13 | 0.98 | 0.98 | 18.7 ±12.4 | 19.0 ±12.7 |
| | SF 3 – 5 | 1486 | 0.80 | 0.82 | 0.82 | 15.4 ±17.9 | 17.5 ±20.4 |
| | SF > 5 | 474 | 8.87 | 1.48 | 0.81 | 30.0 ±31.7 | 14.2 ±23.0 |
| Tower 2 | SF < 3 | 3847 | 0.30 | 0.98 | 0.98 | 46.5 ±25.1 | 47.0 ±25.6 |
| | SF 3 – 5 | 569 | 1.38 | 0.85 | 0.72 | 30.3 ±25.5 | 34.0 ±28.6 |
| | SF > 5 | 212 | 7.94 | 1.60 | 0.76 | 35.6 ±32.5 | 17.3 ±22.7 |

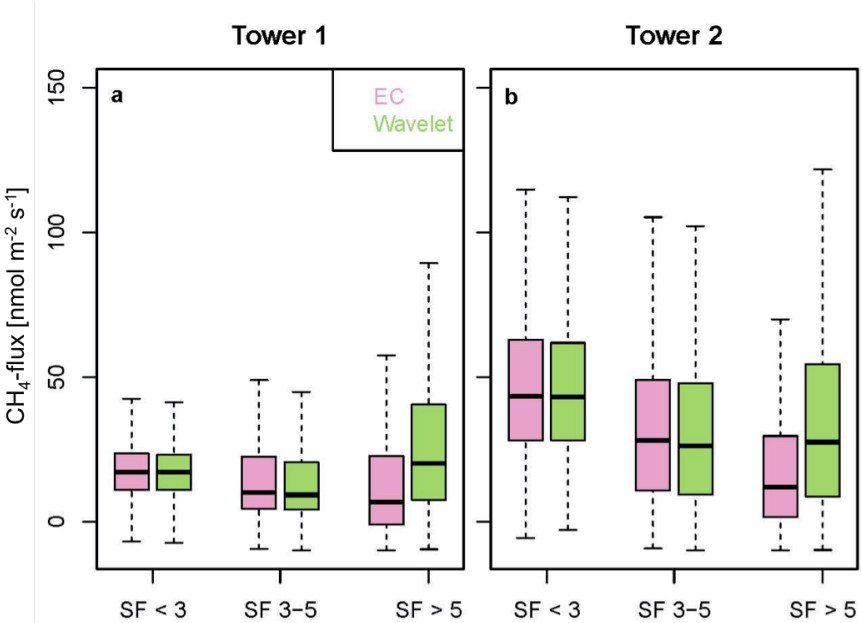

**Figure 1: Median and variability of non-gapfilled methane fluxes based on the EC (purple) and wavelet (green) flux processing methods for different stationarity classes at towers 1 (a) and tower 2 (b). Black horizontal bars give the median, colored boxes indicate the interquartile range covered by the 2nd and 3rd quartile, while whiskers show the minimum and maximum, respectively, flux rates.**

Low stationarity (SF > 5). For this evaluation of fluxes under low stationarity conditions, measured EC fluxes with low quality flags had to be used. Please note that such data would normally have been filtered out during the EC quality control procedure, leaving gaps that would subsequently be filled by gapfilling algorithms. A comparison of methods including such gapfilled data will be presented in the following section, while here the low EC data quality influences the findings. As to be expected, under these circumstances the flux processing methods agree less than under medium or high stationarity conditions, with both slopes and intercepts derived through the OR analysis increasing considerably (Table 3). Also averaged flux rates for the entire study period deviate strongly between methods, with the EC fluxes strongly underestimating the wavelet reference. In comparison to high and medium stationarity conditions, also a wider range of wavelet-based fluxes is found at both towers. These results indicate that non-stationarity flow conditions cause a low bias in the EC-derived methane fluxes in comparison to the wavelet method (Table 3, Figure 1).

### 3.2.2 Influence of flux processing method and gapfilling on flux budgets

To evaluate the impact of discarding portions of an EC-dataset due to low stationarity (SF > 5) in flow conditions, in this section we only used original EC data with medium to high quality, and subsequently filled all gaps with the three different gapfilling algorithms linear interpolation (LI), moving window (MW), and neural network (NN). Since a strong focus is

placed on the evaluation of the gapfilling methods, timestamps with gaps in the wavelet time series, linked to missing data and the range test filter, were subsequently removed also from the gapfilled EC time series to facilitate a direct intercomparison between both methods without having to compare gapfilled to gapfilled values. Accordingly, the resulting flux budgets discussed in the method intercomparison below are not equal to the total methane emissions during the study

period; however, with more than 90 % wavelet data coverage for both towers (Table 2), deviations should be moderate, and overall patterns should be representative. As a reference, filling all gaps in the EC flux time series, at tower 1 seasonal budgets sum up to 2.26, 2.09 and 2.08 gC m$^{-2}$ for LI, MW and ANN respectively, while at tower 2, seasonal budgets are 5.03, 5.09 and 5.02 gC m$^{-2}$ across these three methods.

When integrating the entire dataset, the direct intercomparison of half-hourly fluxes between EC- and wavelet methods

based on OR analyses yields good agreement for tower 2 across gapfilling methods (slope: 1.01 – 1.05; r: 0.88 – 0.89), while at tower 1, a weaker agreement between both flux processing methods after the gapfilling of the EC time series was found (slope: 1.14 – 1.32; r: 0.69 – 0.74). Plotting the frequency distribution of gapfilled flux rates against wavelet results (Figure 2) reveals the important role of the gapfilling performance in this context: for both towers, the reference fluxes provided by the wavelet processing are positively skewed, with a long tail indicating a prominent role of occasional high to very high

methane emissions. The gapfilling algorithms, all of which display comparatively restricted flux ranges, cannot reproduce this distribution, and individual half-hourly flux rates show a poor correlation with the wavelet reference (see also Figure A3 in Appendix D). This applies particularly to the MW and NN approaches, while the flux distribution of the rather simple linear interpolation (LI) at least approximates the positive skewness of the reference. The example of tower 1 demonstrates that this systematic deviation may lead to biases in average flux values produced by the gapfilling methods: in this case,

while wavelet results feature an average methane flux of 30.9±33.4 nmol m$^{-2}$ s$^{-1}$ for those timestamps where EC fluxes were filtered out due to low stationarity, the corresponding gapfilled values in the EC time series had mean flux rates of 24.4±21.2 (-20%, LI), 18.4±8.6 (-41%, MW) and 18.4±9.3 nmol m$^{-2}$ s$^{-1}$ (-41%, NN). On the other hand, at tower 2, in spite of the differences in frequency distributions (Figure 2), smaller shifts in mean flux rates were found, and gapfilled fluxes tended to slightly overestimate the wavelet reference fluxes (see also Figure 4).

For the calculation of long-term methane budgets, the above-mentioned biases in gapfilling results become more important at tower 1, in part also because of the overall higher percentage of gaps compared to tower 2 (Table 2). This is reflected in the fraction of the cumulative methane budget contributed by gapfilled values, which makes up 12 – 15 % at tower 1, but only 6 – 8 % at tower 2 (Table 4). In spite of these deviations, accumulated fluxes for the entire study period are in very good agreement between flux processing methods, and also between gapfilling methods: Flux budgets based on wavelets

sum up to 1.96 gC m$^{-2}$ and 4.56 gC m$^{-2}$ for tower 1 and 2, respectively. Across the three gapfilling approaches, deviations to these reference flux budgets ranged between -0.07 and 0.01 gC (-3.5 – 0.5 %) for tower 1, and between 0.06 and 0.14 gC (1.5 – 3.1%) for tower 2.

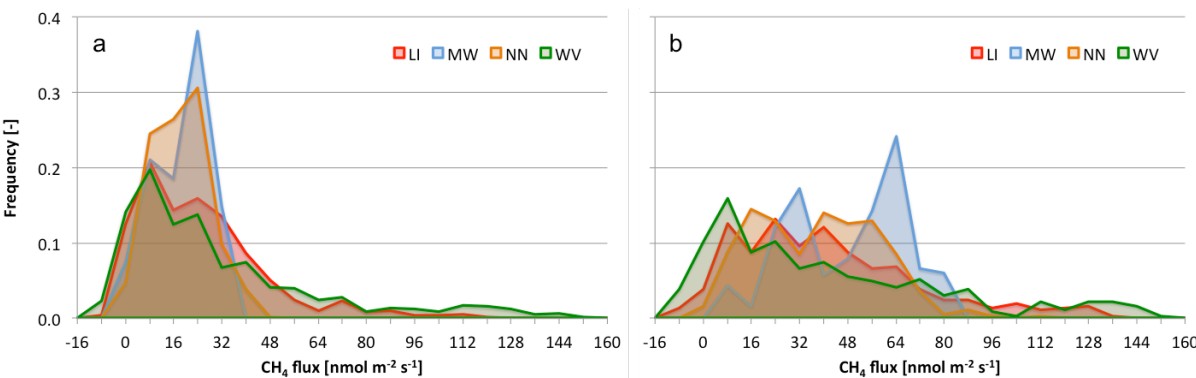

**Figure 2: Frequency distribution of flux rates (a: tower 1; b: tower 2) produced by the three gapfilling approaches (red: linear interpolation (LI); blue: moving window (MW); orange: neural network (NN)), compared against the reference flux values derived from the wavelet raw data processing method (WV, green).**

Table 4: Methane fluxes (FCH₄) summed up for the wavelet method and EC method by applying the three different gapfilling approaches linear interpolation (LI), moving window (MW) and neural network (NN). Note that the same gaps as for the fluxes based on the wavelet method are used for the EC flux time series.

| | Budget | Wavelet | EC_LI | EC_MW | EC_NN |
|---|---|---|---|---|---|
| **Tower 1** | $\sum$ FCH₄ [gC m⁻²] | 1.96 | 1.97 | 1.90 | 1.90 |
| | $\sum$ FCH₄_EC - $\sum$ FCH₄_wavelet [gC m⁻²] | - | 0.01 | -0.07 | -0.07 |
| | $\sum$ gapfilled FCH₄ [gC m⁻²] | 0 | 0.31 | 0.23 | 0.23 |
| | $\sum$ gapfilled FCH₄ / $\sum$ FCH₄_EC [%] | 0 | 15.59 | 12.13 | 12.11 |
| **Tower 2** | $\sum$ FCH₄ [gC m⁻²] | 4.56 | 4.65 | 4.70 | 4.62 |
| | $\sum$ FCH₄_EC - $\sum$ FCH₄_wavelet [gC m⁻²] | - | 0.09 | 0.14 | 0.06 |
| | $\sum$ gapfilled FCH₄ [gC m⁻²] | 0 | 0.32 | 0.37 | 0.30 |
| | $\sum$ gapfilled FCH₄ / $\sum$ FCH₄_EC [%] | 0 | 6.95 | 7.96 | 6.45 |

## 3.3 Analysis of methane emission events

### 3.3.1 Distribution of stationarity classes for different event types

We restricted this analysis to flux data from tower 2, since here the overall higher methane fluxes were measured (see also Section 3.2.1). Similar patterns were found at tower 1 (not shown). The vast majority of 30-minute flux values (5123 cases, or 97%) were categorized as 'No events', i.e. none of the three event types could be detected (Figure 3). This category differs substantially from the 'event' categories regarding the frequency distribution of stability filter (SF) classes: 76 % of cases fell into the high stationarity range (classes 1 & 2), and only 12 % were labeled as low stationarity. The 'detected events' statistics combine 26 half-hourly fluxes from the category 'Peak events', 9 'Updown/Downup events', and 123 'Cluster events'. Across these categories, the percentage of high stationarity data only makes up about 17 % of the dataset,

while the percentage of low stationarity data has been more than doubled to 32 %, compared to the 'no events' category. The majority of cases (~51 %), however, is classified as medium stationarity.

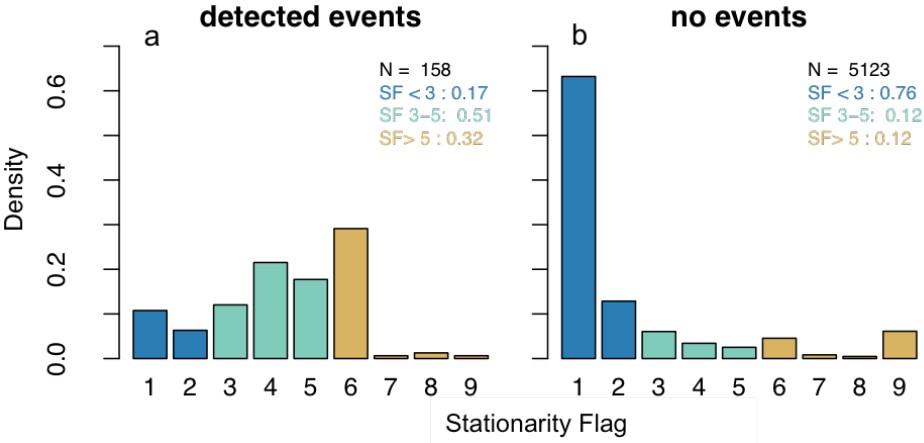

**Figure 3: Stationarity flag frequency distribution of half-hourly timestamps, separating between fluxes that were influenced by events (a) and those where no events were detected (b). Stationarity flags were grouped into the three classes high (dark blue), medium (light blue) and low (light brown) stationarity. The total count of timestamps is slightly above the sum of timestamps available during the study period (5280) due to an occasional occurrence of several events in a single half-hour window.**

### 3.3.2 Methane flux rates during different types of events

Our dataset from tower 2 demonstrates that mean methane flux rates differed between event types (see also Figure 4, similar trends observed at tower 1). Across stationarity categories, average fluxes, where wavelet fluxes were available, were highest during cluster events (wavelet: 52.8 nmol m$^{-2}$ s$^{-1}$, gapfilled EC fluxes ranging between 49.8 – 57.2 nmol m$^{-2}$ s$^{-1}$). In comparison, during peak events flux rates were lower by about 26 % (wavelet: 39.0 nmol m$^{-2}$ s$^{-1}$, gapfilled EC: 36.6 – 41.5 nmol m$^{-2}$ s$^{-1}$), while updown/downup events feature the lowest emissions (wavelet: 21.3 nmol m$^{-2}$ s$^{-1}$, gapfilled EC: 22.4 nmol m$^{-2}$ s$^{-1}$). At times where no events had been detected, wavelet emissions averaged at 44.0 nmol m$^{-2}$ s$^{-1}$, while gapfilled EC fluxes were slightly higher around 44.7 – 45.4 nmol m$^{-2}$ s$^{-1}$.

Comparing the three different stationarity classes, similar patterns emerge across event types, confirming the overall results displayed in Figure 1: During high stationarity, the highest median (Figure 4) and mean flux rates were found across event categories, with wavelet flux rates during peak events as the single exception. Results agree well between processing methods, with no systematic difference observed in either median or mean flux rates. At medium stationarity, mean flux rates are consistently lower than at high stationarity, and the differences in medians as shown in Figure 4 indicate a minor positive offset in flux rates between EC and wavelet methods. At low stationarity, wavelet-derived flux rates are slightly higher again, compared to medium stationarity. EC-based mean fluxes severely underestimate this reference by fractions ranging between -26 – -53 %. Replacing these low quality measurement data with gapfilled values clearly improves the agreement between wavelet and EC-based time series, albeit with a large scatter across methods. For all event types, using any of the three gapfilling algorithms reduces the net offsets to the wavelet-derived fluxes, compared to the original EC data,

with results tending to overestimate the wavelet reference. For LI and MW gapfilling methods, all mean fluxes are higher compared to the wavelets, while NN produces event fluxes lower than this reference, and 'no event' fluxes that are slightly higher. Detailed results are listed in Table A1, Appendix C.

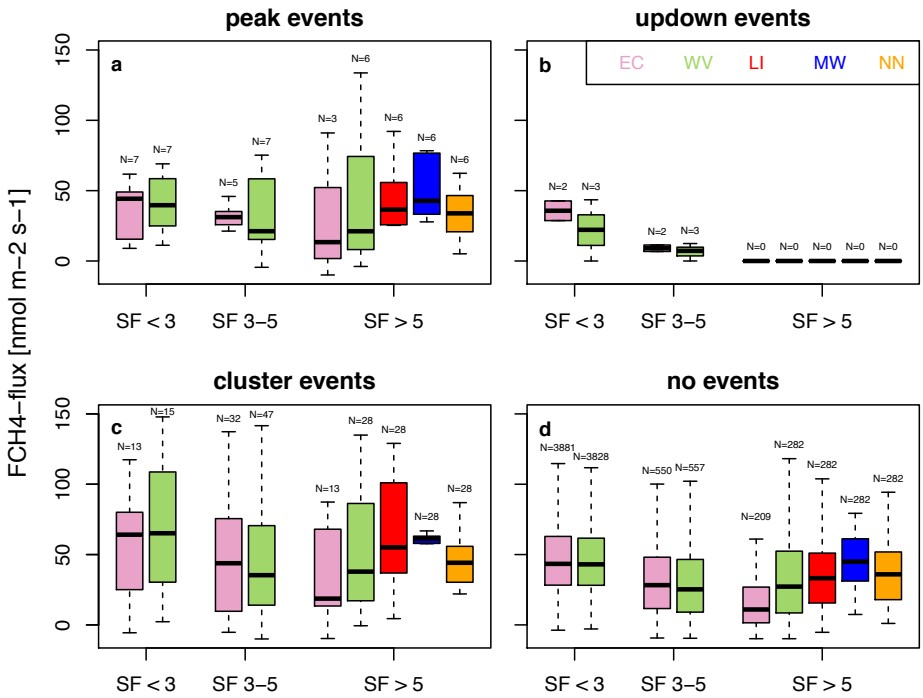

**Figure 4: Methane fluxes based on the EC (purple) and wavelet (green) flux processing method for three stationarity flag (SF) categories during different event types at tower 2. For SF > 5, where methane fluxes based on the EC method would be excluded during the regular post-processing quality control, in addition the values for the three gapfilling methods (red: LI; blue: MW; orange: NN) are shown. For details on graph features, please refer to Figure 2.**

### 3.3.3 Event contribution to methane flux budgets

As to be expected from the low fraction of half-hourly timestamps containing detected events (~3 % at tower 2, see also Figure 3), the total flux budgets are dominated by methane emissions from the 'no events' category. Summed up for tower 2 (Table 5), across the four processing versions (wavelet, EC with three gapfilling approaches) the contributions from events to the total methane budget ranged between 2.5 – 2.8 %. Owing to the dominant fraction of cluster events in those timestamps where events were detected, this event category makes up about 85 % of fluxes influenced by events.

Regarding the role of flow stationarity, the budgets reflect well the distribution of stationarity flags shown above in Figure 3: For the fluxes during 'events', 44 – 51 % of the budget were emitted during medium stationarity, with the remaining flux portions about equally distributed between high and low stationarity. For the 'no events' cases, on the other hand, about 85 % of the total methane emissions can be attributed to high stationarity cases, and only 9 % of the fluxes belong into the medium stationarity category.

Regarding the intercomparison of wavelet and EC-based flux budgets, including the influence of the gapfilling approaches, it needs to be considered that the range test filtered out values at different timestamps between flux processing methods, and the resulting gaps can occur within any stationarity category. Accordingly, the performance of the gapfilling algorithm slightly influenced also the flux budgets for high and medium stationarity, while the biggest impact is found under low

stationarity where results are exclusively based on gapfilling output. Sorting by event type, gapfilled EC flux sums tend to be slightly higher than the wavelet reference, with the exception of NN-budgets for peak and cluster events. Sorting events by stationarity, results summarized in Table 5 indicate that events at high stationarity tend to be underestimated by ~18 %, while medium and low stationarity cases have a high bias (11 % and 5 %, respectively). For 'no events' cases, the gapfilled EC methane budgets have a high bias across stationarity categories and gapfilling algorithms, with only minor flux increases for

high stationarity (~1 %) that increase gradually towards low stationarity. Total flux sums for both 'events' and 'no events' cases are on average overestimated by 2.2 % by the gapfilled EC time series, although with different variability across methods (events: -5.6 – 7.5 %; no events: 1.7 – 3.1 %).

**Table 5: Methane fluxes summed up for the wavelet and EC methods [mgC m$^{-2}$] for tower 2. For EC fluxes, gaps resulting from**
**sorting out low stationarity cases were filled using three different gapfilling approaches (LI: linear interpolation; MW: moving window; NN: neural network). Please note that gaps in the wavelet method were projected to the EC flux time series to ensure a homogeneous database for this method intercomparison.**

| Event type | Stationarity | Wavelet | EC_LI | EC_MW | EC_NN |
|---|---|---|---|---|---|
| **Peak** | All | 16.9 | 17.3 | 17.9 | 15.8 |
| **Updown/downup** | All | 1.8 | 1.9 | 1.9 | 1.9 |
| **Cluster** | All | 102.7 | 111.3 | 106.8 | 96.8 |
| **SUM** | | 121.4 | 130.5 | 126.7 | 114.6 |
| | SF < 3 | 29.9 | 24.0 | 25.0 | 24.4 |
| **All events** | SF 3 – 5 | 53.5 | 61.2 | 59.1 | 58.3 |
| | SF >5 | 37.9 | 45.3 | 42.4 | 31.7 |
| **SUM** | | 121.4 | 130.5 | 126.6 | 114.5 |
| | SF < 3 | 3847 | 3893 | 3891 | 3891 |
| **No events** | SF 3 – 5 | 381 | 401 | 408 | 400 |
| | SF >5 | 207 | 224 | 274 | 218 |
| **SUM** | | 4435 | 4518 | 4573 | 4509 |

# 4 Discussion

## 4.1 Deviations in absolute flux rates between event types and stationarity classes

Mean absolute methane flux rates showed a uniform pattern with respect to the stationarity of the flow (e.g. Figure 1), with fluxes within the highest stationarity class (SF < 3) displaying the highest flux rates. The flux rates under medium stationarity were clearly lowest, while low stationarity ranged somewhere in between the other two classes. Also averaged flux rates for event types (Figure 4) showed some distinctive differences, ranking event types in the order cluster events, no events, peak events, and updown/downup events from high to low average flux rates. These patterns in absolute flux rates, however, strongly depend on the distribution of events and/or stability classes over season and time of day. Therefore, it cannot be ruled out that at least part of the differences between these averaged flux rates have to be attributed to seasonal and/or diurnal variability in methane emissions.

Diurnal variability in flux rates, as e.g. observed at tower 2 within the peak summer season, may particularly alter the comparison of mean flux rates between 'events' and 'no events'. With the majority of events being detected during nighttime (Schaller et al., 2019), higher overall flux rates during the day would mostly raise the 'no events' flux rates. Accordingly, the slightly lower averaged fluxes during 'peak events' (wavelet: 39.0 nmol m$^{-2}$ s$^{-1}$), compared to 'no events' (wavelet: 44.0 nmol m$^{-2}$ s$^{-1}$) may to a large part reflect the time of sampling, rather than an impact of the mechanism of flux release in form of an event on the amount of emitted methane.

## 4.2 Comparison between wavelet- and EC-derived fluxes under different stationarity classes

Excluding gapfilled values from the analysis, we achieved an excellent correlation between wavelet- and EC-derived methane flux rates at high stationarity of the flow. This agreement across processing methods under well-developed atmospheric turbulence, which has been reported before by Schaller et al. (2017; 2019), applies to both the regression analysis of half-hourly fluxes (Table 3) as well as the statistics on averaged flux rates integrated over the study period (see e.g. Figure 1). Given that the assumptions for the application of wavelet flux processing are more relaxed compared to the EC-method, mainly because there is no requirement for stationarity of the flow, wavelet-derived fluxes therefore provide a solid reference for constraining potential biases in EC-fluxes under non-ideal conditions.

Under medium stationarity, mean EC-flux rates are slightly higher than the wavelet reference fluxes at both towers (tower 1: +1.99 nmol m$^{-2}$ s$^{-1}$; tower 2: +0.63 nmol m$^{-2}$ s$^{-1}$). This offset may be linked to the comparatively high flux contribution from half-hourly fluxes influenced by 'events' under this category, which is more than one order of magnitude higher than under high stationarity (see e.g. Table 5, which also includes gapfilled values, however). Disregarding the possible influence of events, even though the differences between flux processing methods are not significant due to the high scatter of flux rates across the entire summer season, a persistent offset in this category will affect the computation of net methane flux budgets, since the overall data quality is still considered high enough that values will not be filtered out during the EC data quality screening.

Our flux processing method intercomparison under low stationarity clearly indicates that EC-derived methane fluxes under such conditions are unreliable, and should be sorted out to ensure plausible results. Mean flux rates for both towers only amounted to slightly more than 50 % of the wavelet reference fluxes, therefore the inclusion of such data into the computation of long-term methane flux budgets would lead to a systematic and potentially severe underestimation of the actual emissions. Since a reliable direct measurement with the EC-method is not reliable, and also gapfilling is associated with considerable uncertainties (see below), wavelet processing holds the potential to provide novel insights into methane exchange processes also under difficult measurement conditions.

## 4.3 Role of gapfilling for EC-derived methane budgets

As demonstrated by the frequency distributions of methane flux rates derived by wavelet processing and three different gapfilling methods (Figure 2), all EC gapfilling approaches tested here cannot capture the full range of natural variability of the methane emissions observed by the reference wavelet fluxes. The wavelet flux distribution indicates that the occurrence of high flux rates, or emission outbursts that may be related to 'events' as further discussed below, are an important element of the methane release dynamics at our study site. These high flux rates, which cause the positive skewness and the long positive tail in the wavelet flux frequency distribution, are at best coarsely approximated by the gapfilling algorithms. The fact that the simplest gapfilling algorithm, linear interpolation, gets closest to a positively skewed flux distribution as provided by the wavelet reference indicates that even sophisticated algorithms such as neural networks have limitations when it comes to capturing the mechanisms that control episodic high methane emissions from wetland ecosystems.

While the uncertainty associated with methane gapfilling produces partly large offsets when comparing individual 30-minute flux rates to the wavelet results, we found the integrated flux over a longer-term study period to be rather stable across gapfilling approaches, and that mean flux rates still agree well with the reference for parts of the dataset. At our tower 2, the gapfilled mean fluxes ranging between $37.2 - 41.3$ nmol m$^{-2}$ s$^{-1}$ agree well with the wavelet mean flux of $37.7$ nmol m$^{-2}$ s$^{-1}$, while at tower 1 the wavelet reference of $30.9$ nmol m$^{-2}$ s$^{-1}$ was clearly underestimated ($18.4 - 24.6$ nmol m$^{-2}$ s$^{-1}$). Based on this finding, we speculate that the decisive factor for the performance of gapfilling algorithms is the mean EC flux during high and medium stationarity, which forms the basis to inform gapfilling algorithms, as well as the diurnal and seasonal gap distributions.

As any other type of model, gapfilling approaches need to be based on reliable statistical and/or process-based algorithms, and in addition need representative training data to produce reliable results. In the tests conducted within the context of this study, none of the three gapfilling algorithms could fully hold up to these standards. For the two simple approaches, linear interpolation and moving window averaging, with no mechanisms available that link fluxes to controls these methods can only rely on the available range of measured fluxes under high to medium stationarity to base their output on. As a consequence, in the absence of process-based algorithms all gapfilling methods are dependent on the distribution of gaps to be filled, and therefore their performance is subject to a certain level of randomness. Regarding the neural network approach, since our example at tower 1 demonstrates that this sophisticated algorithm can produce offsets as large as found for the MW

method, the established links between environmental controls and methane fluxes, which again are based on observations during high or medium stationarity, are not necessarily representative under poorly developed turbulence. This caveat can only be improved through reliable, process-based gapfilling algorithms that do not exclusively focus on biogeochemical aspects, but also incorporate biogeophysical elements such as atmospheric pressure or turbulence conditions into the calculations.

With only up to 11 % of flux values to be filled as gaps resulting from low data quality during our study (Table 2), even a systematic underestimation of reference fluxes by the gapfilling methods of -20 – -41 % at tower 1 did not result in substantial offsets in net methane emissions budgets integrated over the study period (Table 4). This good agreement in net flux budgets may also be linked to the fact that the underestimation of gapfilled values is at least partly balanced by the overestimation of fluxes by the EC-method during medium stationarity. In general, however, it can be expected that the agreement between gapfilled product and reference will significantly deteriorate with an increasing gap fraction within the study dataset. To reduce the associated high uncertainties, wavelet tools as presented herein hold the potential to produce reference datasets under various environmental conditions that can be used to develop, calibrate and test new process-based gapfilling algorithms that are capable to produce reliable results also under low stationarity conditions, i.e. when they are needed most.

## 4.4 Impact of event emissions on methane observations

Our datasets demonstrate that 'event' emissions make up a small but noticeable part of the methane flux time series observed at our Ambolikha observation sites in Northeast Siberia. At tower 2, summed up over the study period of 108 days in summer 2014, about 3 % (158 cases) of half-hourly flux values were affected by events, contributing 2.5 – 2.8 % of the total methane budget emitted during this period. At tower 1 (data not shown), the event fraction was slightly higher (3.7 %, 193 cases), and also the fraction of the total flux affected by events was increased in comparison to tower 2 (3.7 – 5.3 %). Differences between towers are associated with the higher fraction of extreme outliers as detected by the MAD test at tower 1 (Schaller et al., 2019), which may be linked to the fact that mean flux rates at this site are lower, so that emission peaks differ more strongly from the baseline emissions. Overall, these results indicate that, even when completely ignoring the potential presence of such events, regular EC data processing and gapfilling algorithms on average can produce flux rates that are reasonably close to the wavelet fluxes that resolve events (see detailed discussion below). Consequently, for the case study presented herein, the presence of non-stationary methane outburst events did not lead to systematic biases in the EC-based long-term methane budget that go beyond the regular measurement uncertainty.

At tower 2, at times without event occurrence the EC-derived fluxes overestimated the wavelet reference by 1.2 % under high stationarity, and 5.8 % under medium stationarity. Similar offsets were observed at tower 1 (not shown). During events, the overestimation of fluxes under medium stationarity (11 %) approximately matched these biases, while under high stationarity, fluxes tended to be underestimated by 18 %. At tower 1, on the other hand, event fluxes under both stationarity categories were underestimated by 9 – 13 %. With the contributions of total fluxes per stationarity category ranging between

0.7 – 2.8 % across towers, this minor tendency towards underestimating event fluxes did not influence the EC-computed flux budgets considerably.

During low stationarity conditions, all fluxes based on EC-processing will be sorted out, and will subsequently be replaced by gapfilling values, independent whether or not an event was contained in the specific half-hourly window. Therefore, the correspondence between gapfilling results and wavelet reference fluxes was largely identical between 'events' and 'no events' cases at both towers. The influence of events under such circumstances is therefore restricted to the question whether or not event occurrences increase the fraction of detected low stationarity cases, which will be filtered out during quality screening and therefore create gaps. Data summarized in Table 5 show that, for our dataset from tower 2, the relative fraction of cases with low stationarity was ~31 % across half-hourly fluxes that were influenced by events, compared to only 4.7 % for 'no events' cases. This observation indicates that, in general, more events hold the potential to cause more gaps in the flux time series, therefore with more events the gapfilling becomes more important.

Regarding the impact of events on the short-term variability of fluxes, the range of differences between wavelet- and EC-derived 30-minute flux rates is similar for 'events' and 'no events' cases (see Appendix D, Fig. A3); however, while during 'no events' cases a large number of values still shows good correspondence, those cases with substantial deviations from the 1:1 line dominate the method intercomparison for fluxes influenced by events. This is clearly indicated by the root mean square errors (Table A2), which under all stability categories are higher for the events cases. Under high to medium stationarity, the offsets produced by EC-processing appear to be random, therefore the number of events does not seem to introduce a systematic bias into the long-term budget. Still, Figure A3 demonstrates that, particularly for medium stationarity, the EC-derived flux rates influenced by events have a poor quality overall, with RMSE values >40 nmol m$^{-2}$ s$^{-1}$ found for both towers.

Our findings demonstrate that regular eddy-covariance flux processing yields highly reliable results under high stationarity conditions, while for medium to low stationarity, the half-hourly averaged flux rates by the wavelet method should be preferred instead when investigating methane emission dynamics at high temporal resolution. Particularly in the presence of events, individual EC flux rates are associated with a very high uncertainty, and should only be used for the computation of long-term flux budgets. With events often occurring at timescales of only a few minutes, the wavelet flux processing holds the potential to provide new insights into the characteristics of these important elements of the methane cycle, since it facilitates flux computation down to timesteps of one minute without violating underlying theoretical assumptions. As demonstrated already for the decomposition of flux signals from spatially varying source areas (Metzger et al., 2013; Xu et al., 2017), wavelets provide a valuable tool for investigating the statistics of highly irregular emissions, how they can be correlated with environmental conditions, and potentially be resolved by process-based algorithms for gapfilling and/or extrapolation purposes.

# 5 Conclusions

Our study investigated the impact of short-term episodic emission outbursts, so-called event fluxes, on the overall data quality of methane fluxes observed by eddy-covariance towers over a wet tussock tundra ecosystem in Northeast Siberia. We evaluated the EC flux dataset against reference fluxes based on wavelet processing, which are not restricted to stationary flow conditions, and can resolve flux patterns down to timesteps of one minute. The wavelet analysis demonstrates that high methane emission events influenced $3 - 4\,\%$ of the flux observations during our study period, with integrated event emissions contributing $3 - 6\,\%$ to the net methane budget. EC flux data processing tended towards slightly underestimating the wavelet fluxes while events were present, but the net impact on long-term flux budgets is minor in relation to other uncertainties associated with eddy-covariance measurements. For the intercomparison of flux rates at 30-minute timesteps, however, our results demonstrate that the presence of events substantially increases the scatter between wavelet- and EC-derived fluxes, indicating that events introduce additional uncertainty into the EC-results.

A second focus of this study was placed on the evaluation of common gapfilling approaches for EC-derived methane fluxes. Our wavelet-derived fluxes provided an observation-based reference for the fraction of gaps in the EC time series that was created because measurements under low stationarity were filtered out by the data quality assessment protocol. None of the three gapfilling approaches tested herein could reproduce the range of values provided by the wavelet reference, but resulting biases in long-term flux budgets were still minor because of the comparatively small fraction of gaps that needed to be filled in our datasets. The performance of the gapfilling methods appeared to be dependent on the gap distribution, and the ratio of flux rates between the gaps and the remaining dataset. With a profound mechanistic understanding on processes and controls that govern the short-term variability in methane emissions still lacking, the quality of gapfilling products retains a certain level of randomness, therefore systematic biases even over longer timeframes cannot be ruled out particularly for datasets that contain a higher gap fraction as the ones used in our study.

Our findings demonstrate that wavelet analyses hold the potential to enhance our understanding in methane exchange processes between terrestrial ecosystems and the atmosphere. With excellent agreement between wavelet- and EC-derived fluxes demonstrated under ideal turbulence conditions, wavelet fluxes facilitate to quantify biases in EC-datasets linked to non-ideal conditions, e.g. medium to low stationarity of the flow. Moreover, the provision of observationally-based reference fluxes at times when the EC-method produces data gaps can support the development of novel process-based modeling algorithms that are representative for a wider range of environmental conditions, which can be employed e.g. for the improvement of gapfilling algorithms. Finally, the option to resolve fluxes down to temporal resolutions of one minute facilitates new insights into the intermittent nature of methane emissions, and its impact on the quality of methane flux observations.

## Acknowledgements

The authors would like to thank Prof. Dr. Thomas Foken (University of Bayreuth) for his comments to an earlier version of the manuscript. This work was supported through funding by the European Commission (PAGE21 project, FP7-ENV-2011, Grant Agreement No. 282700; PerCCOM project, FP7-PEOPLE-2012-CIG, Grant Agreement No. PCIG12-GA-201-333796; INTAROS project, H2020-BG-09-2016, Grant Agreement No. 727890, Nunataryuk project, H2020-BG-11-2016/17, Grant Agreement No. 773421), the German Ministry of Education and Research (CarboPerm project, Grant No. 03G0836G; KoPf project, Grant No. 03F0764D), and the AXA Research Fund (PDOC_2012_W2 campaign, ARF fellowship M. Göckede). Furthermore the German Academic Exchange Service (DAAD) provided financial support for the travel expenses. The authors appreciate the contribution of staff members of the Northeast Scientific Station in Chersky for facilitating the field experiments.

## Software availability

The software scripts that execute the wavelet-based flux data processing can be made available by the authors upon request.

## Author contributions

MG was responsible for study conception and supervision. All authors contributed to development of the methodology and formal data analysis, with computation largely carried out by FK (eddy covariance) and CS (wavelets). All authors contributed to underlying fieldwork. MG wrote the initial manuscript, with contributions by FK. All authors contributed to reviewing the manuscript text, and editing the final manuscript version.

## Appendix A: Wavelet approach to calculate turbulent fluxes

The following description of the wavelet method is a slightly shortened version of the description provided by Schaller et al. (2017), which is the companion paper introducing the methodology that the presented study is based upon. It has been included here again to facilitate an easier overview on the procedure, without having to read other manuscripts. For more
details, please refer to Schaller et al. (2017).

A continuous wavelet transform of a discrete time series $x(t)$ can be written as convolution of $x(t)$,

$$T(a,b) = \int_{-\infty}^{\infty} x(t) \cdot \psi_{a,b}^{*}(t)dt \qquad \text{(Eq. A1)}$$

where $T(a,b)$ is the wavelet coefficient and $\Psi_{a,b}(t)$ is referred to as wavelet function

$$\Psi_{a,b}(t) = \frac{1}{\sqrt{a}} \cdot \Psi\left(\frac{t-b}{a}\right). \qquad \text{(Eq. A2)}$$

The wavelet $\Psi$ requires a dilation parameter $a$, which controls the scale of the wavelet and thus the current frequency of interest, and a translation parameter $b$ that indicates the temporal position of the wavelet in the time series. For as complex-valued wavelet, the conjugate $\Psi_{a,b}^{*}(t)$ denoted by a star sign is used.

As mentioned in the main text above, this study used the complex-valued Morlet wavelet for quantification of flux rates, and the Mexican Hat wavelet for the exact localization of $CH_4$ emission events (see Schaller et al., 2017 for more details). The
expression $T^2(a,b)$ across all times and scales provides the total energy of the time series. The average of the wavelet scalogram $|T^2(a,b)|$ is used to obtain the wavelet spectrum (Torrence and Compo, 1998)

$$E_x(j) = \frac{\delta t}{C_\delta} \cdot \frac{1}{N} \cdot \sum_{n=0}^{N-1} |T^2(a,b)| \qquad \text{(Eq. A3)}$$

over a given number $N$ of values in the time series, taking the time step $\delta t$ and a wavelet-specific reconstruction factor $C_\delta$ into account. From this it is now possible to obtain the global variance of the time series, $\sigma_x^2$, by integrating over all scales $j = 0$
to $J$

$$\sigma_x^2 = \frac{\delta t}{C_\delta} \cdot \frac{\delta j}{N} \cdot \sum_{n=0}^{N-1} \sum_{j=0}^{J} \frac{|T^2(a,b)|}{a(j)} \qquad \text{(Eq. A4)}$$

with $\delta_j$ referring to the spacing between discrete scales and $J$ being the maximum number of scales.

For two simultaneously recorded time series $x(t)$ and $y(t)$ the wavelet cross spectrum can now be obtained in analogy to Eq. (A3) as

$$E_{xy}(j) = \frac{\delta t}{C_\delta} \cdot \frac{1}{N} \cdot \sum_{n=0}^{N-1} \left[T_x(a,b) \cdot T_y^{*}(a,b)\right], \qquad \text{(Eq. A5)}$$

where $T_y^{*}(a,b)$ denotes the complex conjugate of the wavelet transform of the second time series $y(t)$ (Hudgins et al., 1993). Summing up over all scales yields the covariance (Stull, 1988)

$$\overline{x'y'} = \frac{\delta t}{C_\delta} \cdot \frac{\delta j}{N} \cdot \sum_{n=0}^{N-1} \sum_{j=0}^{J} \frac{\left[T_x(a,b) \cdot T_y^{*}(a,b)\right]}{a(j)} \qquad \text{(Eq. A6)}$$

for the chosen averaging interval. If the chosen time series $x$ and $y$ are the vertical wind velocity $w$ and a corresponding gas
concentration $c$, the flux $\overline{w'c'}$ can be calculated now using Equation (A6).

**Appendix B: Seasonal and diurnal pattern of flux rates and data gaps**

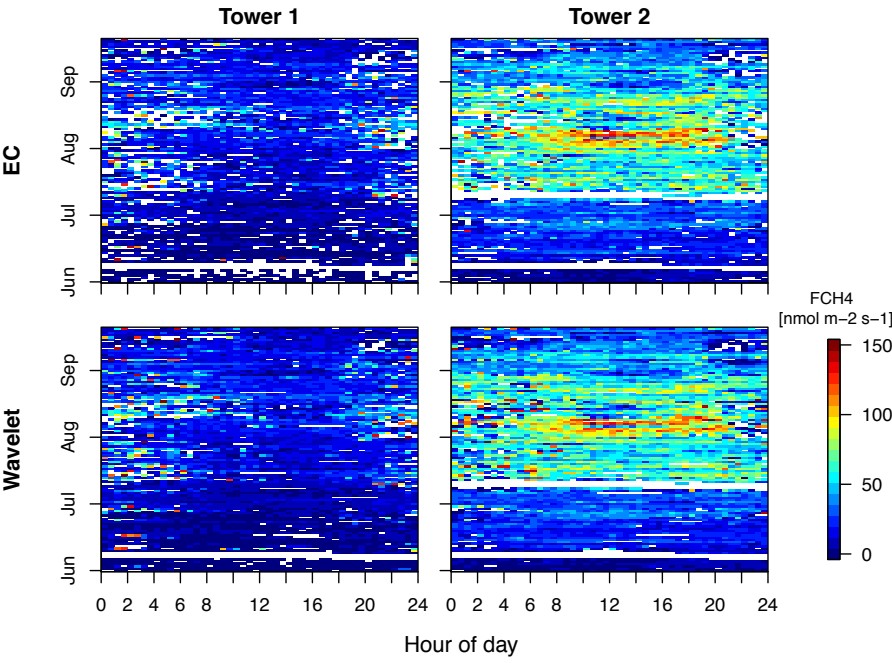

**Figure A1: Fingerprint plots showing the diurnal distribution of flux rates and gaps (white) for both towers and processing methods.**

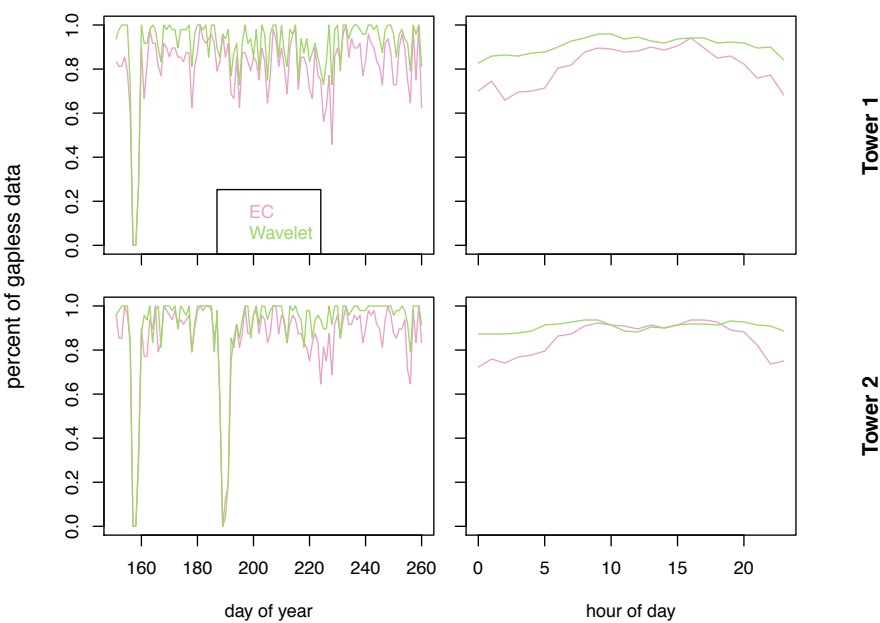

**Figure A2: Seasonal (left) and diurnal (right) distribution of data availability for wavelet- (green lines) and EC-derived (purple lines) methane fluxes.**

**Appendix C: Mean methane flux rates for different event categories**

**Table A1: Mean methane fluxes for the wavelet method and EC method, split into three stationarity categories. For the lowest stationarity, in addition to measured EC values, model results by the three different gapfilling approaches linear interpolation (LI), moving window (MW) and neuronal network (NN) are given. Absolute flux values are given in cells with white background, while grey shading indicates flux differences between EC- and wavelet processing, where numbers in brackets give the percentage deviation.**

| | Flux calculation method | Peak [nmol m$^{-2}$ s$^{-1}$] | Updown/downup [nmol m$^{-2}$ s$^{-1}$] | Cluster [nmol m$^{-2}$ s$^{-1}$] | No event [nmol m$^{-2}$ s$^{-1}$] |
|---|---|---|---|---|---|
| **High stationarity** | Wavelet | 40.97 | 32.83 | 68.94 | 46.52 |
| | EC | 34.87 | 35.71 | 55.16 | 47.03 |
| | EC-Wavelet | -6.1 (-14%) | 2.88 (8.8%) | -13.78 (-20%) | 0.51 (1.1%) |
| **Medium stationarity** | Wavelet | 34.2 | 9.81 | 47.22 | 31.67 |
| | EC | 31.87 | 9.07 | 50.06 | 33.1 |
| | EC-Wavelet | -2.33 (-6.8%) | -0.74 (-7.5%) | 2.84 (6.0%) | 1.43 (4.5%) |
| **Low stationarity** | Wavelet | 42.42 | NA | 53.59 | 34.05 |
| | EC_measured | 31.5 | NA | 32.71 | 16.07 |
| | EC_gapfilled LI | 45.34 | NA | 65.17 | 36.8 |
| | EC_gapfilled MW | 50.28 | NA | 59.43 | 45.02 |
| | EC_gapfilled NN | 33.75 | NA | 45.22 | 35.72 |
| | EC_measured-Wavelet | -10.92 (-26%) | NA | -20.88 (-39%) | -17.98 (-53%) |
| | EC_gapfilled LI-Wavelet | 2.92 (6.9%) | NA | 11.58 (22%) | 2.75 (8.1%) |
| | EC_gapfilled MW-Wavelet | 7.86 (19%) | NA | 5.84 (11%) | 10.97 (32%) |
| | EC_gapfilled NN-Wavelet | -8.67 (-20%) | NA | -8.37 (-16%) | 1.67 (4.9%) |

**Appendix D: Comparison of methane flux rates at half-hourly resolution between methods**

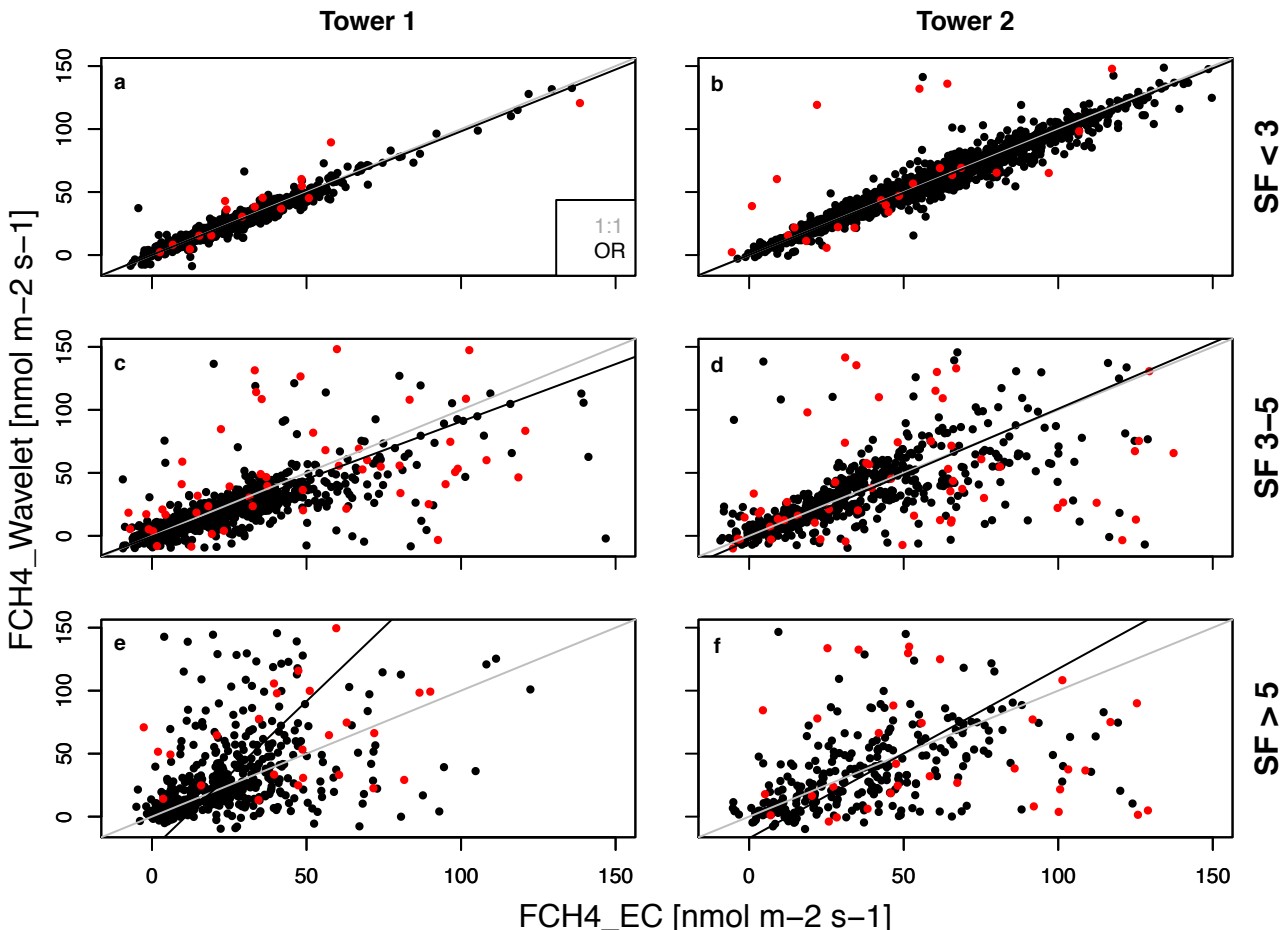

**Figure A3: Impact of events on the direct intercomparison of half-hourly flux rates between wavelet and eddy-covariance processing methods, sorted by tower and stationarity flag (SF) category. The displayed dataset includes gapfilled data, where linear interpolation was used to fill gaps for the EC method under low stationarity. Fluxes influenced by events are plotted in red, while 'no events' cases are black. The thin grey line gives the 1:1 line, the black line the fit of the orthogonal regression (OR) analysis.**

**Table A2: Deviations between 30-minute averaged fluxes based on wavelet- and EC-processing, expressed as the root mean square error [nmol m$^{-2}$ s$^{-1}$]. Time series include gapfilled values from linear interpolation for the EC time series.**

| Stationarity | Tower 1 | | Tower 2 | |
|---|---|---|---|---|
| | No events | Events | No events | Events |
| High (SF < 3) | 2.64 | 11.89 | 4.54 | 34.04 |
| Medium (SF 3 − 5) | 12.11 | 40.75 | 21.97 | 48.61 |
| Low (SF > 5) | 27.67 | 42.06 | 28.12 | 60.76 |

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

# Quantifying the impact of emission outbursts and non-stationary flow on eddy covariance CH$_4$ flux measurements using wavelet techniques

5  Mathias Göckede[1,*], Fanny Kittler[1], Carsten Schaller[1,2]

[1]Max Planck Institute for Biogeochemistry, Jena, Germany
[2]now: University of Münster, Institute of Landscape Ecology, Climatology Research Group, Münster, Germany

*Correspondence to*: Mathias Göckede (mathias.goeckede@bgc-jena.mpg.de)

**Abstract.** Methane flux measurements by the eddy-covariance technique are subject to large uncertainties, particularly linked to the partly highly intermittent nature of methane emissions. Outbursts of high methane emissions, termed event fluxes, hold the potential to introduce systematic biases into derived methane budgets, since under such conditions the assumption of stationarity of the flow is violated. In this study, we investigate the net impact of this effect by comparing
eddy-covariance fluxes against a wavelet-derived reference that is not negatively influenced by non-stationarity. Our results demonstrate that methane emission events influenced 3-4 % of the flux measurements, and did not lead to systematic biases in methane budgets for the analyzed summer season; however, the presence of events substantially increased uncertainties in short-term flux rates. The wavelet results provided an excellent reference to evaluate the performance of three different gapfilling approaches for eddy-covariance methane fluxes, and we show that none of them could reproduce the range of
observed flux rates. The integrated performance of the gapfilling methods for the longer-term dataset varied between the two eddy-covariance towers involved in this study, and we show that gapfilling remains a large source of uncertainty linked to limited insights into the mechanisms governing the short-term variability in methane emissions. With the capability to broaden our observational methane flux database to a wider range of conditions, including the direct resolution of short term variability at the order of minutes, wavelet-derived fluxes hold the potential to generate new insight into methane exchange
processes with the atmosphere, and therefore also improve our understanding of the underlying processes.

## 1 Introduction

The eddy covariance (EC) technique, a well-established method for the direct quantification of turbulent surface–atmosphere exchange processes (Aubinet et al., 2012), can provide valuable information on current CH$_4$ flux rates between various types of ecosystems and the atmosphere (e.g. Taylor et al., 2018; Rößger et al., 2019; Tuovinen et al., 2019), including insights
into processes and controls (e.g. Pirk et al., 2016; Kittler et al., 2017b; Neumann et al., 2019) that can be used to improve future projections. However, the data quality of EC measurements depends strongly on the adherence to several theoretical assumptions such as e.g. steady-state conditions and horizontal homogeneity (Foken, 2017), which frequently limits data

**M G 5/27/2019 22:04**
**Deleted:** The Arctic, with its abundant wetlands (Andresen et al., 2017) and enormous soil carbon pools (Hugelius et al., 2014), is of high relevance for the global methane (CH$_4$) budget (Saunois et al., 2016), particularly regarding the potential for substantial increases in CH$_4$ emissions from this region under future climate scenarios (IPCC, 2013).

**M G 5/27/2019 22:04**
**Deleted:** insights

**M G 5/27/2019 22:04**
**Deleted:** into

**M G 5/27/2019 22:04**
**Deleted:** Arctic

**M G 5/27/2019 22:05**
**Deleted:** a sparse observational coverage currently limits insights from the available Arctic EC database: Arctic permafrost ecosystems are often located in remote regions that are difficult to access, and harsh climate conditions pose special requirements to EC instrumentation and power supply (Goodrich et al., 2016; Kittler et al., 2017a). Moreover,

**M G 5/27/2019 22:05**
**Deleted:** further

**M G 5/27/2019 22:05**
**Deleted:** reducing

availability. In case of methane fluxes, particularly a potential violation of the required steady-state conditions linked to episodic outbursts from wetland sources (Schaller et al., 2019) may lead to low flux data quality, and therefore can substantially increase the gap fraction in quality filtered EC time series.

One potential mechanism for such high methane emission events is so-called ebullition (e.g. Kwon et al., 2017; Peltola et al., 2018; Männistö et al., 2019), i.e. periodic bubble outgassing with a typical length of seconds to minutes. Even though such emissions are part of the natural flux signal, and should therefore be accounted for when accumulating longer-term budgets of methane exchange, in the context of EC data processing and quality assessment these events are likely to be discarded during the quality screening of raw data, or they may be incorrectly handled by the data processing algorithms. In both cases, the natural high flux event would be incorrectly accounted for, potentially introducing systematic biases into methane fluxes and budgets (Baldocchi et al., 2012).

Spatial heterogeneity in the emission patterns of methane surrounding the flux tower (e.g. Rey-Sanchez et al., 2019) may also lead to pronounced variability in the observed $CH_4$ flux time series (Tuovinen et al., 2019). Particularly for wetland ecosystems, ecosystem characteristics such as inundation level or vegetation composition may vary at finest spatial scales (Muster et al., 2012; McEwing et al., 2015), creating microsite variability with strong gradients in methane emissions. Also at landscape (Peltola et al., 2015) to regional scales (Davidson et al., 2016), spatial variability in landscape characteristics may have a strong influence on the captured flux signal. For flux towers situated in such structured areas, an emission spike in the $CH_4$ flux time series can also be created by a temporary shift of the field of view of the sensors from a low flux region into a high flux region and back (e.g. Korrensalo et al., 2018). As outlined for the ebullition fluxes above, depending on the exact nature of the spike the flux signal may be misinterpreted by the eddy-covariance processing software.

Since 'outburst events' in methane fluxes are in many cases flagged as non-stationary conditions, and therefore discarded as low-quality data, the assessment of the net impact of this effect needs to consider what will happen to the resulting gaps in the quality-filtered EC time series. Gaps are a common feature in eddy-covariance time series, resulting e.g. from power failures, instrument malfunctioning, or low data quality linked to the violation of the above-mentioned theoretical assumptions (e.g. Foken et al., 2004). If they can be filled with a reliable, unbiased algorithm, additional gaps would not pose a major problem. For $CO_2$, several of such well-established frameworks are available (e.g. Reichstein et al., 2005; Moffat et al., 2007), allowing to generate continuous time series for the assessment of long-term flux budgets. In contrast, for $CH_4$ fluxes no consensus on a gap-filling method has yet emerged within the EC-community. Several studies succeeded in establishing data-driven links between $CH_4$ fluxes and environmental conditions such as e.g. peat/soil temperature, friction velocity or water table (e.g. Wille et al., 2008; Zona et al., 2009; Jackowicz-Korczynski et al., 2010) using both linear and non-linear functional relationships. Other approaches include gap interpolation (e.g. Rinne et al., 2007; Tagesson et al., 2012), process-based modeling (Forbrich et al., 2011) or artificial neural networks (e.g. Dengel et al., 2013). Even though these different approaches have been shown to perform well in case studies, a solution that has been proven to be uniformly applicable is lacking, therefore large uncertainties are still associated with $CH_4$ gapfilling.

M G 5/27/2019 22:05
Deleted: .

As an alternative to the regular eddy-covariance raw data processing, the flux calculations can also be performed based on the wavelet method by analyzing frequency patterns in the underlying time series of winds and scalars (Collineau and Brunet, 1993b, a). In contrast to the eddy-covariance method, the wavelet method is not restricted by the same set of theoretical assumptions, and in particular no steady-state conditions are required (e.g. Daubechies, 1990). Wavelets have
5 been demonstrated to be a powerful tool for quantifying turbulent fluxes (Mauder et al., 2007; Thomas and Foken, 2007). The ability to calculate turbulent fluxes for periods as short as one minute has been proven very valuable for attributing flux variability to environmental controls, both based on aircraft campaigns (Metzger et al., 2013) and stationary tower measurements within a heterogeneous landscape (Xu et al., 2017). Moreover, wavelet techniques have been applied to improve the frequency correction with the eddy-covariance method (Nordbo and Katul, 2013). A direct comparison between
10 fluxes processed with the wavelet and eddy-covariance method, respectively, found an excellent agreement between both methods for EC data of highest quality (Schaller et al., 2017).

Here, we quantify the net impact of failing to resolve methane outburst events with the EC method, comparing both short-term emission patterns and longer-term flux budgets to a reference flux product derived with wavelet methods. The presented study is closely linked to two recently published manuscripts (Schaller et al., 2017; 2019) that demonstrate that
fluxes during such outburst events, with timescales at the order of only a few minutes, can be precisely quantified using the wavelet method, while the coarser temporal resolution of the EC method normally fails to resolve these details while aggregating over 30 minutes. In this follow-up study, we determine systematic offsets between both methods, and the specific role that different types of short-term outburst events play in this context. Since many non-stationary events were leading to data gaps in the EC flux time series, we placed a specific focus on evaluating the performance of different gap-
filling algorithms to fill these gaps. Overall, our study aims at evaluating the effect of non-stationary conditions on the long-term methane flux budgets, with a special focus placed on systematic biases introduced by either flux processing approach or chosen gap-filling method.

## 2 Material and Methods

### 2.1 Site description

The Ambolikha research site (Göckede et al., 2017), located on a floodplain of the Kolyma River approximately 18 km south of the town of Chersky, northeast Russia, is underlain by continuous permafrost and characterized as wet tussock tundra dominated by tussock-forming *Carex appendiculata* and *lugens* and *Eriophorum angustifolium* (Corradi et al., 2005; Kwon et al., 2016). Alluvial mineral soils (silty clay) are topped by an organic peat layer (0.15–0.20 m), with some of the organic material also present in deeper layers following cryoturbation (Corradi et al., 2005; Merbold et al., 2009). Averaged for the
period 1960 – 2009, the mean annual air temperature was -11°C and the average annual precipitation summed up to 197 mm (Göckede et al., 2017). Vegetation height was ~ 0.7 m during the peak of the growing season, reached around the beginning of August.

Data were collected from two eddy-covariance towers situated about 600 m apart, both elevated ~6 m above sea level. While one measurement system was placed within a drainage ditch system (tower 1, 68.61 °N and 161.34 °E, FLUXNET code RU-Che), therefore capturing fluxes that represent a patch of tundra affected by a lowered water table, the second control measurement system (tower 2, 68.62 °N and 161.35 °E, RU-Ch2) measures natural exchange conditions unaffected by the hydrological disturbance. For this study, a dataset covering the period June 01 to September 18, 2014, was used.

## 2.2 Instrument setup

Both flux towers mentioned above in Section 2.1 were equipped with the same instrumentation, including a sonic anemometer (uSonic-3 Scientific, 5 W heating, METEK GmbH, Elmshorn, DE) at the tower top (at heights of 4.9 m and 5.1 m for drained and control tower, respectively) and a closed-path greenhouse gas analyzer for $CH_4/CO_2/H_2O$ (FGGA, Los Gatos Research Inc., CA, USA). Ambient air was drawn by an external vacuum pump (membrane pump, N940, KNF, 13 L min$^{-1}$ under ambient pressure) from an inlet placed next to the sonic anemometer (vertical sensor separation: 0.30 m) through a heated and insulated sampling line (Eaton Synflex decabon with 6.2 mm inner diameter and a length of 16 m and 13 m for drained and control tower, respectively). The acquisition of high frequency (20 Hz) raw data was handled by the software package EDDYMEAS (Kolle and Rebmann, 2007) on a local computer at the field site.

Ancillary meteorological data were collected at 10 s intervals from both towers and stored as 10-minute averages on a data logger (CR3000, Campbell Scientific, UT, USA). Acquired parameters include e.g. air temperature and humidity, air pressure, precipitation, or soil temperatures. Low-frequency meteorological data underwent a thorough data quality control screening, and subsequently were averaged to 30 minutes (see Kittler et al. (2016) for details).

## 2.3 Raw data processing

We based the raw data processing to obtain fluxes from the collected high frequency data on two different methods:

1. The eddy-covariance raw data processing uses the software package TK3 (Mauder and Foken, 2015). When applied in stand-alone mode, this tool implements all required conversions, corrections, and quality assessment procedures (Foken et al., 2012; Fratini and Mauder, 2014). Details on the TK3 implementation on the Ambolikha datasets are provided by Kittler et al. (2016; 2017a).

2. The second flux processing method (Schaller et al., 2017; 2019) is based on wavelet analysis and uses the sinusoidal and complex valued Morlet wavelet transform for flux quantification. The Morlet wavelet provides an excellent resolution in the frequency domain and can be used to analyze atmospheric turbulence (e.g. Strunin and Hiyama, 2004; Thomas and Foken, 2005). Since this study focused on comparing eddy covariance- and wavelet-derived fluxes, the temporal integration of the wavelet method was chosen to closely match the eddy covariance method (30 min); however, due to the decomposition in time and frequency domain the averaging intervals could not match perfectly, and an averaging interval of 33 min for the wavelet method was used. A detailed description of the wavelet method, the wavelet transform and the corresponding flux data processing can be found in appendix A and in Schaller et al. (2017).

In the context of the presented study, in a first processing step both methods were applied to produce continuous time series of uncorrected half-hour fluxes of methane. In a subsequent processing stage, the results provided by both methods underwent the same flux correction procedure by the TK3 software package, including 2D coordinate rotation of the wind field, cross-wind correction (Liu et al., 2001), and correction for losses in the high-frequency range (Moore, 1986).

The eddy covariance post-processing quality control is commonly based on the analysis of stationary and well-developed turbulence conditions (e.g. Foken et al., 2004; 2012). When applied for wavelet fluxes, the test for stationarity can be dropped, since wavelet flux data quality is not compromised by non-stationary conditions (see above). The development of the turbulence is investigated based on the concept of flux-variance similarity (Wyngaard et al., 1971) via the so-called integral turbulence characteristics (ITC, Foken and Wichura, 1996). A low data quality rating by the ITC can e.g. be caused

by stable atmospheric stratification that suppresses turbulent motions. Data stationarity is tested by comparing signal covariance at different averaging intervals (e.g. 5-minutes vs. 30-minutes, Foken and Wichura, 1996). In this context, effects such as e.g. spikes in the signal, abrupt changes of the signal level, or intermittent turbulence may trigger low flux data quality. We grouped eddy covariance fluxes into different categories (Table 1) based on their stationarity flag (SF) ratings. Fluxes outside the range -10 nmol $m^{-2}$ $s^{-1}$ < $CH_4$ flux < 150 nmol $m^{-2}$ $s^{-1}$ (based on the 2.5 % and 97.5 % quantiles for high

and medium quality $CH_4$ fluxes from tower 2) were sorted out during the post-processing.

Gaps of the eddy covariance time series were filled with three different methods. A linear interpolation (LI) represents the simplest method. The mean of a 10-day moving window (MW) centered to the gap was used for a better representation of the seasonality. These two methods were chosen since they do not require a sophisticated tool, and can thus be easily applied. Finally, a neuronal network approach (NN, Dengel et al., 2013) represents a more sophisticated gapfilling algorithm,

filling gaps based on prevailing environmental conditions.

To assess the agreement between EC and wavelet fluxes, a regression analysis was applied. With flux data of both methods being subject to uncertainties, no independent variable could be identified. Thus, in place of ordinary least-square regression, an orthogonal regression  (OR, linear model II regression) was used with the R-package "lmodel2" (Legendre, 2014) and Pearson's correlation coefficients (r) are given. OR is particularly suited for the comparison of time series that are both

subject to errors of about the same order of magnitude (e.g. Foken, 2017).

**Table 1: Quality flag categories based on the stationarity rating of the eddy covariance flux data. The definition of quality categories follows the scheme proposed by Sabbatini et al. (2018), which is based on stationarity tests developed by Foken et al. (2004; 2012), but uses stricter thresholds to separate categories.**

| Quality | Stationarity flag (SF) | Range of differences [%]* |
|---------|------------------------|---------------------------|
| High | < 3 | 0 – 30 |
| Medium | 3 – 5 | 31 – 100 |
| Low | >5 | > 100 |

*Difference [%] between the covariances calculated over 30 minutes and calculated as a average of six 5-minute covariances (for details please refer to Foken and Wichura, 1996).

### 2.4 Event characterization

The characterization of high methane emission event types differentiated within the context of this study is based on a wavelet approach using the Mexican Hat wavelet. In contrast to the Morlet wavelet, which we used to precisely quantify flux rates due to its excellent localization in the frequency domain, the Mexican Hat wavelet has a very good localization in the time domain, therefore facilitated an exact localization of single events. Event periods resolved at minute intervals were identified by the median absolute deviation (MAD, e.g. Hoaglin et al., 2000) test followed by an additional manual adjustment. Events were separated into the three categories introduced by Schaller et al. (2019):

1. Peak events: This simple event starts from a baseline flux level, monotonically changes towards a peak or plateau, and subsequently monotonically changes back to the baseline level again.

2. Updown/Downup events: Similar to two connected peak events with opposite sign, after reaching a first peak the fluxes overshoot the baseline level to reach a second peak in the opposite direction before approaching the baseline again. An up-down event indicates a positive peak followed by a negative one, for a down-up event the sequence would be vice versa.

3. Cluster events: Prolonged periods containing numerous high methane emission events were labeled as cluster events. Such periods showed a distinctive pattern of high emissions, compared to the baseline fluxes before and after the event, but did not display the clearly defined peak structures as defined above.

### 3 Results

### 3.1 Data coverage and overall quality flags

Of the 5280 half-hourly flux values that would provide continuous data coverage within the study period June 01 to September 18, 2014, about 3.4 % or 6 % of the eddy fluxes were either missing or discarded as lowest data quality for tower 1 and tower 2, respectively (Table 2). For the wavelet datasets, missing flux values had a slightly higher percentage compared to the EC dataset, since a required 3-hour window of continuous data focusing on the current timestamp

broadened the window of missing fluxes around every gap in the raw data. From the remaining data, a further 11.8 % (tower 1) or 6.6 % (tower 2) were discarded during the EC quality control procedure as low quality, in all cases linked to non-stationary flow conditions. Since the wavelet method does not require stationarity, no additional gaps due to low data quality occurred. For both methods, the subsequent range test (see Section 2.3 for details) filtered out another 2 – 5 % of data that was assigned high to medium quality. Taken together, for each combination of tower and processing method, more than 80 % of the fluxes remained after quality screening. Compared to the wavelets, this percentage is lower by about 7 % for the EC method, linked to the requirement of stationary flow conditions.

Regarding the distribution of gaps over time, no seasonal patterns were found for both towers and both flux processing methods, so each part of the study period received about equal data coverage. With respect to diurnal patterns in gap distribution, EC data display a higher gap fraction during the night, compared to daytime data coverage. This imbalance is most pronounced for tower 1 (see also Appendix B, Figures A1 & A2). No such diurnal patterns in gap distribution were found within the wavelet flux time series, and also no systematic differences between both towers were found for this method. Taken together, wavelet flux data processing provides a better overall data coverage, i.e. less data gaps have to be filled to replace unreliable measurements flagged as low quality. Also, the equal distribution of gaps between day and night supports an improved performance of gap-filling algorithms. Since gapfilled fluxes are associated with higher uncertainties than measured fluxes, this indicates that wavelet data processing holds the potential to produce more robust flux budgets.

**Table 2: Gap fraction within the dataset used for this study, separated by flux processing method and tower position.**

| Gaps [%] | Tower 1 | | Tower 2 | |
|---|---|---|---|---|
| | EC | Wavelet | EC | Wavelet |
| missing data, or lowest quality in raw dataset | 3.43 | 4.13 | 5.95 | 6.93 |
| low quality flag during post-processing | 11.80 | - | 6.59 | - |
| range test flag for remaining medium/high quality data | 3.39 | 5.09 | 2.22 | 2.52 |
| TOTAL SUM | 18.62 | 9.22 | 14.75 | 9.45 |

**3.2 Flux data quality analysis**

**3.2.1 Comparison of non-gapfilled methane fluxes under different stationarity conditions**

In this section, we compare measured methane flux rates between EC and wavelet methods, with the intention to derive the dependence of differences between methods on the stationarity of the underlying flow conditions. This analysis excludes gapfilled results. A comparison between methods focusing on the derivation of long-term flux budgets, which include also the gapfilled values, will be presented in the following Section 3.2.2. For both flux processing methods, higher methane emissions under all quality and stationarity conditions are observed at tower 2 (see also Figure 1), which features a higher fraction of inundated areas in its footprint in comparison to tower 1. At tower 2, for both flux processing methods, flux rates

display a pronounced increase in mid-July, leading to a peak in August and a subsequent decrease at the beginning of September, a pattern that follows the general seasonal trends in soil temperatures. During these times of increased methane emissions, a diurnal cycle with higher flux rates during daytime is observed (see also Figure A1), while at tower 1, for both flux processing methods no seasonal or diurnal cycle was observed.

High stationarity (SF < 3). Under highly stationary flow conditions, we found an excellent agreement between half-hourly flux rates derived with EC and wavelet flux processing, respectively. A direct comparison shows that both methods produce highly correlated $CH_4$-fluxes throughout the spectrum of absolute values, a fact that is confirmed by a orthogonal regression analysis (Wavelet = Intercept + Slope · EC) that produces slopes close to 1, intercepts close to 0 nmol m$^{-2}$ s$^{-1}$, and correlation coefficients of 0.98 for both towers (Table 3). Averaging data under high stationarity for the entire study period yields flux

rates that are only marginally higher for the EC method, compared to the wavelet reference (Table 3, Figure 1).

Medium stationarity (SF 3 – 5). The correlation between half-hourly flux rates derived with both processing methods is reduced under medium stationary flow conditions, compared to the high stationarity. This observation is confirmed by the OR analysis, which produces coefficients that deviate stronger from the ideal targets as shown above for high stationarity (Table 3). Mean flux rates are reduced in comparison to highly stationary conditions, and positive offsets between fluxes

derived by the EC method and the wavelet method, respectively, are higher for both towers (Table 3, Figure 1).

**Table 3: Statistical coefficients of an orthogonal regression analysis (Wavelet = Intercept + Slope · EC) and mean flux rates for the two processing methods separated by stationarity classes (SF<3 as high, SF 3 – 5 as medium, and SF>5 as low stationarity).**

| | Data | N | Intercept [nmol m$^{-2}$ s$^{-1}$] | Slope | r | Mean flux rate [nmol m$^{-2}$ s$^{-1}$] Wavelet | EC |
|---|---|---|---|---|---|---|---|
| **Tower 1** | SF < 3 | 2728 | 0.13 | 0.98 | 0.98 | 18.7 ±12.4 | 19.0 ±12.7 |
| | SF 3 – 5 | 1486 | 0.80 | 0.82 | 0.82 | 15.4 ±17.9 | 17.5 ±20.4 |
| | SF > 5 | 474 | 8.87 | 1.48 | 0.81 | 30.0 ±31.7 | 14.2 ±23.0 |
| **Tower 2** | SF < 3 | 3847 | 0.30 | 0.98 | 0.98 | 46.5 ±25.1 | 47.0 ±25.6 |
| | SF 3 – 5 | 569 | 1.38 | 0.85 | 0.72 | 30.3 ±25.5 | 34.0 ±28.6 |
| | SF > 5 | 212 | 7.94 | 1.60 | 0.76 | 35.6 ±32.5 | 17.3 ±22.7 |

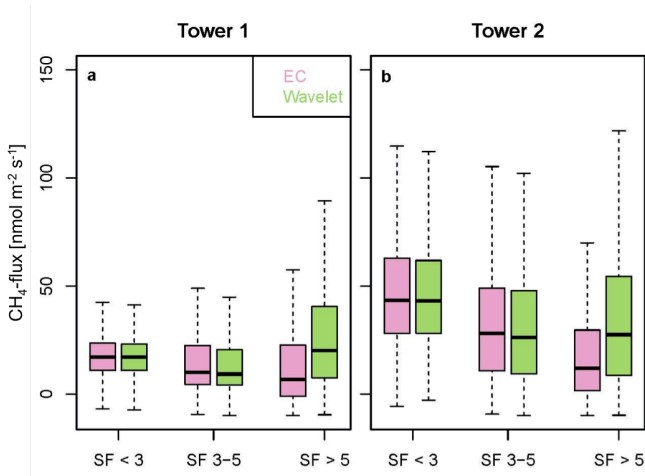

**Figure 1: Median and variability of non-gapfilled methane fluxes based on the EC (purple) and wavelet (green) flux processing methods for different stationarity classes at towers 1 (a) and tower 2 (b). Black horizontal bars give the median, colored boxes indicate the interquartile range covered by the 2nd and 3rd quartile, while whiskers show the minimum and maximum, respectively, flux rates.**

Low stationarity (SF > 5). For this evaluation of fluxes under low stationarity conditions, measured EC fluxes with low quality flags had to be used. Please note that such data would normally have been filtered out during the EC quality control procedure, leaving gaps that would subsequently be filled by gapfilling algorithms. A comparison of methods including such gapfilled data will be presented in the following section, while here the low EC data quality influences the findings. As to be expected, under these circumstances the flux processing methods agree less than under medium or high stationarity conditions, with both slopes and intercepts derived through the OR analysis increasing considerably (Table 3). Also averaged flux rates for the entire study period deviate strongly between methods, with the EC fluxes strongly underestimating the wavelet reference. In comparison to high and medium stationarity conditions, also a wider range of wavelet-based fluxes is found at both towers. These results indicate that non-stationarity flow conditions cause a low bias in the EC-derived methane fluxes in comparison to the wavelet method (Table 3, Figure 1).

**3.2.2 Influence of flux processing method and gapfilling on flux budgets**

To evaluate the impact of discarding portions of an EC-dataset due to low stationarity (SF > 5) in flow conditions, in this section we only used original EC data with medium to high quality, and subsequently filled all gaps with the three different gapfilling algorithms linear interpolation (LI), moving window (MW), and neural network (NN). Since a strong focus is

placed on the evaluation of the gapfilling methods, timestamps with gaps in the wavelet time series, linked to missing data and the range test filter, were subsequently removed also from the gapfilled EC time series to facilitate a direct intercomparison between both methods without having to compare gapfilled to gapfilled values. Accordingly, the resulting flux budgets discussed in the method intercomparison below are not equal to the total methane emissions during the study period; however, with more than 90 % wavelet data coverage for both towers (Table 2), deviations should be moderate, and overall patterns should be representative. As a reference, filling all gaps in the EC flux time series, at tower 1 seasonal budgets sum up to 2.26, 2.09 and 2.08 gC m$^{-2}$ for LI, MW and ANN respectively, while at tower 2, seasonal budgets are 5.03, 5.09 and 5.02 gC m$^{-2}$ across these three methods.

When integrating the entire dataset, the direct intercomparison of half-hourly fluxes between EC- and wavelet methods based on OR analyses yields good agreement for tower 2 across gapfilling methods (slope: 1.01 – 1.05; r: 0.88 – 0.89), while at tower 1, a weaker agreement between both flux processing methods after the gapfilling of the EC time series was found (slope: 1.14 – 1.32; r: 0.69 – 0.74). Plotting the frequency distribution of gapfilled flux rates against wavelet results (Figure 2) reveals the important role of the gapfilling performance in this context: for both towers, the reference fluxes provided by the wavelet processing are positively skewed, with a long tail indicating a prominent role of occasional high to very high methane emissions. The gapfilling algorithms, all of which display comparatively restricted flux ranges, cannot reproduce this distribution, and individual half-hourly flux rates show a poor correlation with the wavelet reference (see also Figure A3 in Appendix D). This applies particularly to the MW and NN approaches, while the flux distribution of the rather simple linear interpolation (LI) at least approximates the positive skewness of the reference. The example of tower 1 demonstrates that this systematic deviation may lead to biases in average flux values produced by the gapfilling methods: in this case, while wavelet results feature an average methane flux of 30.9±33.4 nmol m$^{-2}$ s$^{-1}$ for those timestamps where EC fluxes were filtered out due to low stationarity, the corresponding gapfilled values in the EC time series had mean flux rates of 24.4±21.2 (-20%, LI), 18.4±8.6 (-41%, MW) and 18.4±9.3 nmol m$^{-2}$ s$^{-1}$(-41%, NN). On the other hand, at tower 2, in spite of the differences in frequency distributions (Figure 2), smaller shifts in mean flux rates were found, and gapfilled fluxes tended to slightly overestimate the wavelet reference fluxes (see also Figure 4).

For the calculation of long-term methane budgets, the above-mentioned biases in gapfilling results become more important at tower 1, in part also because of the overall higher percentage of gaps compared to tower 2 (Table 2). This is reflected in the fraction of the cumulative methane budget contributed by gapfilled values, which makes up 12 – 15 % at tower 1, but only 6 – 8 % at tower 2 (Table 4). In spite of these deviations, accumulated fluxes for the entire study period are in very good agreement between flux processing methods, and also between gapfilling methods: Flux budgets based on wavelets sum up to 1.96 gC m$^{-2}$ and 4.56 gC m$^{-2}$ for tower 1 and 2, respectively. Across the three gapfilling approaches, deviations to these reference flux budgets ranged between -0.07 and 0.01 gC (-3.5 – 0.5 %) for tower 1, and between 0.06 and 0.14 gC (1.5 – 3.1%) for tower 2.

M G 6/11/2019 10:45
**Deleted:** Table 2

M G 6/11/2019 10:45
**Deleted:** Figure 2

M G 5/29/2019 20:52
**Deleted:** C

M G 6/11/2019 10:45
**Deleted:** Figure 2

M G 6/11/2019 10:45
**Deleted:** Figure 4

M G 6/11/2019 10:45
**Deleted:** Table 2

M G 6/11/2019 10:45
**Deleted:** Table 4

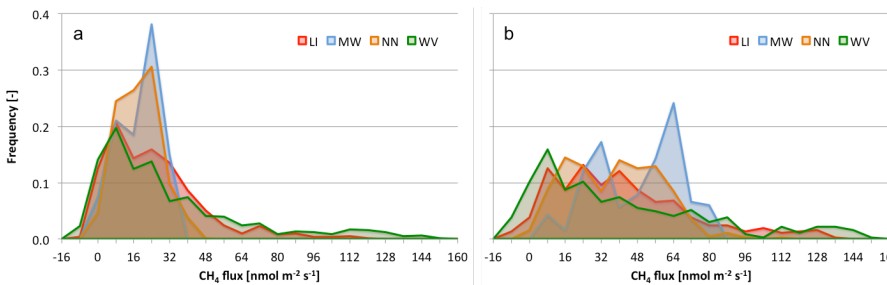

**Figure 2: Frequency distribution of flux rates (a: tower 1; b: tower 2) produced by the three gapfilling approaches (red: linear interpolation (LI); blue: moving window (MW); orange: neural network (NN)), compared against the reference flux values derived from the wavelet raw data processing method (WV, green).**

**Table 4: Methane fluxes (FCH$_4$) summed up for the wavelet method and EC method by applying the three different gapfilling approaches linear interpolation (LI), moving window (MW) and neural network (NN). Note that the same gaps as for the fluxes based on the wavelet method are used for the EC flux time series.**

| | Budget | Wavelet | EC_LI | EC_MW | EC_NN |
|---|---|---|---|---|---|
| **Tower 1** | $\sum$ FCH$_4$ [gC m$^{-2}$] | 1.96 | 1.97 | 1.90 | 1.90 |
| | $\sum$ FCH$_4$_EC - $\sum$ FCH$_4$_wavelet [gC m$^{-2}$] | - | 0.01 | -0.07 | -0.07 |
| | $\sum$ gapfilled FCH$_4$ [gC m$^{-2}$] | 0 | 0.31 | 0.23 | 0.23 |
| | $\sum$ gapfilled FCH$_4$ / $\sum$ FCH$_4$_EC [%] | 0 | 15.59 | 12.13 | 12.11 |
| **Tower 2** | $\sum$ FCH$_4$ [gC m$^{-2}$] | 4.56 | 4.65 | 4.70 | 4.62 |
| | $\sum$ FCH$_4$_EC - $\sum$ FCH$_4$_wavelet [gC m$^{-2}$] | - | 0.09 | 0.14 | 0.06 |
| | $\sum$ gapfilled FCH$_4$ [gC m$^{-2}$] | 0 | 0.32 | 0.37 | 0.30 |
| | $\sum$ gapfilled FCH$_4$ / $\sum$ FCH$_4$_EC [%] | 0 | 6.95 | 7.96 | 6.45 |

### 3.3 Analysis of methane emission events

#### 3.3.1 Distribution of stationarity classes for different event types

We restricted this analysis to flux data from tower 2, since here the overall higher methane fluxes were measured (see also Section 3.2.1). Similar patterns were found at tower 1 (not shown). The vast majority of 30-minute flux values (5123 cases, or 97%) were categorized as 'No events', i.e. none of the three event types could be detected (Figure 3). This category differs substantially from the 'event' categories regarding the frequency distribution of stability filter (SF) classes: 76 % of cases fell into the high stationarity range (classes 1 & 2), and only 12 % were labeled as low stationarity. The 'detected events' statistics combine 26 half-hourly fluxes from the category 'Peak events', 9 'Updown/Downup events', and 123 'Cluster events'. Across these categories, the percentage of high stationarity data only makes up about 17 % of the dataset,

while the percentage of low stationarity data has been more than doubled to 32 %, compared to the 'no events' category. The majority of cases (~51 %), however, is classified as medium stationarity.

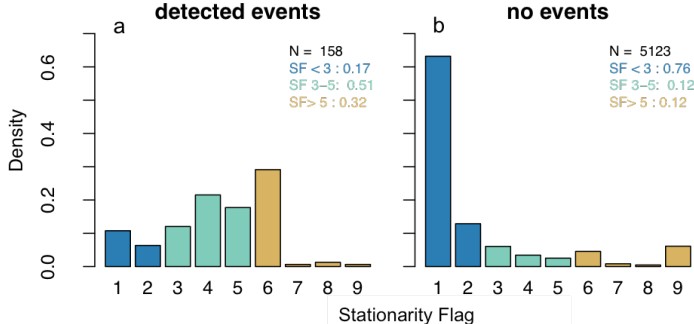

**Figure 3: Stationarity flag frequency distribution of half-hourly timestamps, separating between fluxes that were influenced by events (a) and those where no events were detected (b). Stationarity flags were grouped into the three classes high (dark blue), medium (light blue) and low (light brown) stationarity. The total count of timestamps is slightly above the sum of timestamps available during the study period (5280) due to an occasional occurrence of several events in a single half-hour window.**

### 3.3.2 Methane flux rates during different types of events

Our dataset from tower 2 demonstrates that mean methane flux rates differed between event types (see also Figure 4, similar trends observed at tower 1). Across stationarity categories, average fluxes, where wavelet fluxes were available, were highest during cluster events (wavelet: 52.8 nmol m$^{-2}$ s$^{-1}$, gapfilled EC fluxes ranging between 49.8 – 57.2 nmol m$^{-2}$ s$^{-1}$). In comparison, during peak events flux rates were lower by about 26 % (wavelet: 39.0 nmol m$^{-2}$ s$^{-1}$, gapfilled EC: 36.6 – 41.5 nmol m$^{-2}$ s$^{-1}$), while updown/downup events feature the lowest emissions (wavelet: 21.3 nmol m$^{-2}$ s$^{-1}$, gapfilled EC: 22.4 nmol m$^{-2}$ s$^{-1}$). At times where no events had been detected, wavelet emissions averaged at 44.0 nmol m$^{-2}$ s$^{-1}$, while gapfilled EC fluxes were slightly higher around 44.7 – 45.4 nmol m$^{-2}$ s$^{-1}$.

Comparing the three different stationarity classes, similar patterns emerge across event types, confirming the overall results displayed in Figure 1. During high stationarity, the highest median (Figure 4) and mean flux rates were found across event categories, with wavelet flux rates during peak events as the single exception. Results agree well between processing methods, with no systematic difference observed in either median or mean flux rates. At medium stationarity, mean flux rates are consistently lower than at high stationarity, and the differences in medians as shown in Figure 4 indicate a minor positive offset in flux rates between EC and wavelet methods. At low stationarity, wavelet-derived flux rates are slightly higher again, compared to medium stationarity. EC-based mean fluxes severely underestimate this reference by fractions ranging between -26 – -53 %. Replacing these low quality measurement data with gapfilled values clearly improves the agreement between wavelet and EC-based time series, albeit with a large scatter across methods. For all event types, using any of the three gapfilling algorithms reduces the net offsets to the wavelet-derived fluxes, compared to the original EC data,

with results tending to overestimate the wavelet reference. For LI and MW gapfilling methods, all mean fluxes are higher compared to the wavelets, while NN produces event fluxes lower than this reference, and 'no event' fluxes that are slightly higher. Detailed results are listed in Table A1, Appendix C.

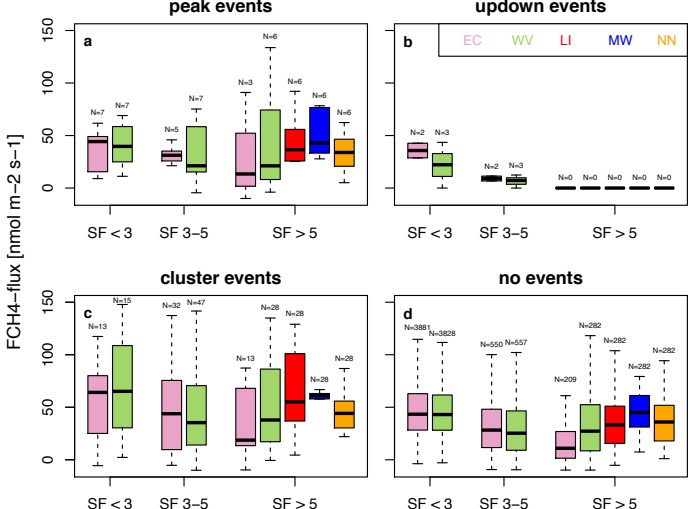

**Figure 4: Methane fluxes based on the EC (purple) and wavelet (green) flux processing method for three stationarity flag (SF) categories during different event types at tower 2. For SF > 5, where methane fluxes based on the EC method would be excluded during the regular post-processing quality control, in addition the values for the three gapfilling methods (red: LI; blue: MW; orange: NN) are shown. For details on graph features, please refer to Figure 2.**

### 3.3.3 Event contribution to methane flux budgets

As to be expected from the low fraction of half-hourly timestamps containing detected events (~3 % at tower 2, see also Figure 3), the total flux budgets are dominated by methane emissions from the 'no events' category. Summed up for tower 2 (Table 5), across the four processing versions (wavelet, EC with three gapfilling approaches) the contributions from events to the total methane budget ranged between 2.5 – 2.8 %. Owing to the dominant fraction of cluster events in those timestamps where events were detected, this event category makes up about 85 % of fluxes influenced by events.

Regarding the role of flow stationarity, the budgets reflect well the distribution of stationarity flags shown above in Figure 3: For the fluxes during 'events', 44 – 51 % of the budget were emitted during medium stationarity, with the remaining flux portions about equally distributed between high and low stationarity. For the 'no events' cases, on the other hand, about 85 % of the total methane emissions can be attributed to high stationarity cases, and only 9 % of the fluxes belong into the medium stationarity category.

M G 5/29/2019 20:52
**Deleted:** B

M G 6/11/2019 10:45
**Deleted: Figure 2**

M G 6/11/2019 10:45
**Deleted:** Figure 3

M G 6/11/2019 10:45
**Deleted:** Table 5

M G 6/11/2019 10:45
**Deleted:** Figure 3

Regarding the intercomparison of wavelet and EC-based flux budgets, including the influence of the gapfilling approaches, it needs to be considered that the range test filtered out values at different timestamps between flux processing methods, and the resulting gaps can occur within any stationarity category. Accordingly, the performance of the gapfilling algorithm slightly influenced also the flux budgets for high and medium stationarity, while the biggest impact is found under low

stationarity where results are exclusively based on gapfilling output. Sorting by event type, gapfilled EC flux sums tend to be slightly higher than the wavelet reference, with the exception of NN-budgets for peak and cluster events. Sorting events by stationarity, results summarized in Table 5 indicate that events at high stationarity tend to be underestimated by ~18 %, while medium and low stationarity cases have a high bias (11 % and 5 %, respectively). For 'no events' cases, the gapfilled EC methane budgets have a high bias across stationarity categories and gapfilling algorithms, with only minor flux increases for

high stationarity (~1 %) that increase gradually towards low stationarity. Total flux sums for both 'events' and 'no events' cases are on average overestimated by 2.2 % by the gapfilled EC time series, although with different variability across methods (events: -5.6 – 7.5 %; no events: 1.7 – 3.1 %).

**Table 5: Methane fluxes summed up for the wavelet and EC methods [mgC m$^{-2}$] for tower 2. For EC fluxes, gaps resulting from**
**sorting out low stationarity cases were filled using three different gapfilling approaches (LI: linear interpolation; MW: moving window; NN: neural network). Please note that gaps in the wavelet method were projected to the EC flux time series to ensure a homogeneous database for this method intercomparison.**

| Event type | Stationarity | Wavelet | EC_LI | EC_MW | EC_NN |
|---|---|---|---|---|---|
| **Peak** | All | 16.9 | 17.3 | 17.9 | 15.8 |
| **Updown/downup** | All | 1.8 | 1.9 | 1.9 | 1.9 |
| **Cluster** | All | 102.7 | 111.3 | 106.8 | 96.8 |
| **SUM** | | 121.4 | 130.5 | 126.7 | 114.6 |
| | SF < 3 | 29.9 | 24.0 | 25.0 | 24.4 |
| **All events** | SF 3 – 5 | 53.5 | 61.2 | 59.1 | 58.3 |
| | SF >5 | 37.9 | 45.3 | 42.4 | 31.7 |
| **SUM** | | 121.4 | 130.5 | 126.6 | 114.5 |
| | SF < 3 | 3847 | 3893 | 3891 | 3891 |
| **No events** | SF 3 – 5 | 381 | 401 | 408 | 400 |
| | SF >5 | 207 | 224 | 274 | 218 |
| **SUM** | | 4435 | 4518 | 4573 | 4509 |

## 4 Discussion

### 4.1 Deviations in absolute flux rates between event types and stationarity classes

Mean absolute methane flux rates showed a uniform pattern with respect to the stationarity of the flow (e.g. Figure 1), with fluxes within the highest stationarity class (SF < 3) displaying the highest flux rates. The flux rates under medium stationarity were clearly lowest, while low stationarity ranged somewhere in between the other two classes. Also averaged flux rates for event types (Figure 4) showed some distinctive differences, ranking event types in the order cluster events, no events, peak events, and updown/downup events from high to low average flux rates. These patterns in absolute flux rates, however, strongly depend on the distribution of events and/or stability classes over season and time of day. Therefore, it cannot be ruled out that at least part of the differences between these averaged flux rates have to be attributed to seasonal and/or diurnal variability in methane emissions.

Diurnal variability in flux rates, as e.g. observed at tower 2 within the peak summer season, may particularly alter the comparison of mean flux rates between 'events' and 'no events'. With the majority of events being detected during nighttime (Schaller et al., 2019), higher overall flux rates during the day would mostly raise the 'no events' flux rates. Accordingly, the slightly lower averaged fluxes during 'peak events' (wavelet: 39.0 nmol m$^{-2}$ s$^{-1}$), compared to 'no events' (wavelet: 44.0 nmol m$^{-2}$ s$^{-1}$) may to a large part reflect the time of sampling, rather than an impact of the mechanism of flux release in form of an event on the amount of emitted methane.

### 4.2 Comparison between wavelet- and EC-derived fluxes under different stationarity classes

Excluding gapfilled values from the analysis, we achieved an excellent correlation between wavelet- and EC-derived methane flux rates at high stationarity of the flow. This agreement across processing methods under well-developed atmospheric turbulence, which has been reported before by Schaller et al. (2017; 2019), applies to both the regression analysis of half-hourly fluxes (Table 3) as well as the statistics on averaged flux rates integrated over the study period (see e.g. Figure 1). Given that the assumptions for the application of wavelet flux processing are more relaxed compared to the EC-method, mainly because there is no requirement for stationarity of the flow, wavelet-derived fluxes therefore provide a solid reference for constraining potential biases in EC-fluxes under non-ideal conditions.

Under medium stationarity, mean EC-flux rates are slightly higher than the wavelet reference fluxes at both towers (tower 1: +1.99 nmol m$^{-2}$ s$^{-1}$; tower 2: +0.63 nmol m$^{-2}$ s$^{-1}$). This offset may be linked to the comparatively high flux contribution from half-hourly fluxes influenced by 'events' under this category, which is more than one order of magnitude higher than under high stationarity (see e.g. Table 5, which also includes gapfilled values, however). Disregarding the possible influence of events, even though the differences between flux processing methods are not significant due to the high scatter of flux rates across the entire summer season, a persistent offset in this category will affect the computation of net methane flux budgets, since the overall data quality is still considered high enough that values will not be filtered out during the EC data quality screening.

M G 6/11/2019 10:45
**Deleted:** Figure 1

M G 6/11/2019 10:45
**Deleted:** Figure 4

M G 5/29/2019 21:18
**Deleted:** As an example, the high average cluster event flux rates may be linked to a preferred occurrence of cluster events at times/dates when higher flux rates are prevalent, while the finding does not necessarily demonstrate that the mechanism behind cluster events also triggers an enhancement in methane emissions.

M G 6/11/2019 10:45
**Deleted:** Table 3

M G 6/11/2019 10:45
**Deleted:** Figure 1

M G 5/29/2019 21:20
**Deleted:** These findings demonstrate that, as long as the theoretic requirements for both methods are fulfilled, no systematic biases between processing methods exist that need to be considered for a method intercomparison.

M G 6/11/2019 10:45
**Deleted:** Table 5

M G 5/29/2019 21:23
**Deleted:** it appears that even minor deviations from ideal flow conditions hold the potential to systematically disturb EC flux computation, e.g. through the assignment of incorrect mean values for vertical wind speed and scalar concentrations during the Reynolds decomposition (e.g. Aubinet et al., 2012). E

Our flux processing method intercomparison under low stationarity clearly indicates that EC-derived methane fluxes under such conditions are unreliable, and should be sorted out to ensure plausible results. Mean flux rates for both towers only amounted to slightly more than 50 % of the wavelet reference fluxes, therefore the inclusion of such data into the computation of long-term methane flux budgets would lead to a systematic and potentially severe underestimation of the

actual emissions. Since a reliable direct measurement with the EC-method is not reliable, and also gapfilling is associated with considerable uncertainties (see below), wavelet processing holds the potential to provide novel insights into methane exchange processes also under difficult measurement conditions.

**4.3 Role of gapfilling for EC-derived methane budgets**

As demonstrated by the frequency distributions of methane flux rates derived by wavelet processing and three different

gapfilling methods (Figure 2), all EC gapfilling approaches tested here cannot capture the full range of natural variability of the methane emissions observed by the reference wavelet fluxes. The wavelet flux distribution indicates that the occurrence of high flux rates, or emission outbursts that may be related to 'events' as further discussed below, are an important element of the methane release dynamics at our study site. These high flux rates, which cause the positive skewness and the long positive tail in the wavelet flux frequency distribution, are at best coarsely approximated by the gapfilling algorithms. The

fact that the simplest gapfilling algorithm, linear interpolation, gets closest to a positively skewed flux distribution as provided by the wavelet reference indicates that even sophisticated algorithms such as neural networks have limitations when it comes to capturing the mechanisms that control episodic high methane emissions from wetland ecosystems.

While the uncertainty associated with methane gapfilling produces partly large offsets when comparing individual 30-minute flux rates to the wavelet results, we found the integrated flux over a longer-term study period to be rather stable across

gapfilling approaches, and that mean flux rates still agree well with the reference for parts of the dataset. At our tower 2, the gapfilled mean fluxes ranging between $37.2 - 41.3$ nmol m$^{-2}$ s$^{-1}$ agree well with the wavelet mean flux of $37.7$ nmol m$^{-2}$ s$^{-1}$, while at tower 1 the wavelet reference of $30.9$ nmol m$^{-2}$ s$^{-1}$ was clearly underestimated ($18.4 - 24.6$ nmol m$^{-2}$ s$^{-1}$). Based on this finding, we speculate that the decisive factor for the performance of gapfilling algorithms is the mean EC flux during high and medium stationarity, which forms the basis to inform gapfilling algorithms, as well as

the diurnal and seasonal gap distributions.

As any other type of model, gapfilling approaches need to be based on reliable statistical and/or process-based algorithms, and in addition need representative training data to produce reliable results. In the tests conducted within the context of this study, none of the three gapfilling algorithms could fully hold up to these standards. For the two simple approaches, linear interpolation and moving window averaging, with no mechanisms available that link fluxes to controls these methods can

only rely on the available range of measured fluxes under high to medium stationarity to base their output on. As a consequence, in the absence of process-based algorithms all gapfilling methods are dependent on the distribution of gaps to be filled, and therefore their performance is subject to a certain level of randomness. Regarding the neural network approach, since our example at tower 1 demonstrates that this sophisticated algorithm can produce offsets as large as found for the MW

---

M G 5/29/2019 21:24

**Deleted:** At the same time, with non-zero flux rates under low stationarity conditions that can exceed those under high and medium stationarity conditions, the wavelet method demonstrates that there is still a substantial flux contribution during periods of non-stationarity.

M G 5/29/2019 21:24

**Deleted:** of such fluxes

M G 6/11/2019 10:45

**Deleted:** Figure 2

M G 6/11/2019 12:01

**Deleted:** s

M G 6/11/2019 12:01

**Deleted:** are

M G 5/29/2019 21:32

**Moved (insertion) [1]**

M G 5/29/2019 21:32

**Deleted:** Since particularly the MW approach, which by default tends to smooth out extreme values, produces a large offset to the wavelet reference at tower 1, we speculate that at this site a dominant fraction of the gaps in the methane time series under low stationarity conditions is made up by high emission events that systematically exceed the flux rates observed under high and medium stationarity.

method, the established links between environmental controls and methane fluxes, which again are based on observations during high or medium stationarity, are not necessarily representative under poorly developed turbulence. This caveat can only be improved through reliable, process-based gapfilling algorithms that do not exclusively focus on biogeochemical aspects, but also incorporate biogeophysical elements such as atmospheric pressure or turbulence conditions into the calculations.

With only up to 11 % of flux values to be filled as gaps resulting from low data quality during our study (Table 2), even a systematic underestimation of reference fluxes by the gapfilling methods of -20 – -41 % at tower 1 did not result in substantial offsets in net methane emissions budgets integrated over the study period (Table 4). This good agreement in net flux budgets may also be linked to the fact that the underestimation of gapfilled values is at least partly balanced by the overestimation of fluxes by the EC-method during medium stationarity. In general, however, it can be expected that the agreement between gapfilled product and reference will significantly deteriorate with an increasing gap fraction within the study dataset. To reduce the associated high uncertainties, wavelet tools as presented herein hold the potential to produce reference datasets under various environmental conditions that can be used to develop, calibrate and test new process-based gapfilling algorithms that are capable to produce reliable results also under low stationarity conditions, i.e. when they are needed most.

### 4.4 Impact of event emissions on methane observations

Our datasets demonstrate that 'event' emissions make up a small but noticeable part of the methane flux time series observed at our Ambolikha observation sites in Northeast Siberia. At tower 2, summed up over the study period of 108 days in summer 2014, about 3 % (158 cases) of half-hourly flux values were affected by events, contributing 2.5 – 2.8 % of the total methane budget emitted during this period. At tower 1 (data not shown), the event fraction was slightly higher (3.7 %, 193 cases), and also the fraction of the total flux affected by events was increased in comparison to tower 2 (3.7 – 5.3 %). Differences between towers are associated with the higher fraction of extreme outliers as detected by the MAD test at tower 1 (Schaller et al., 2019), which may be linked to the fact that mean flux rates at this site are lower, so that emission peaks differ more strongly from the baseline emissions. Overall, these results indicate that, even when completely ignoring the potential presence of such events, regular EC data processing and gapfilling algorithms on average can produce flux rates that are reasonably close to the wavelet fluxes that resolve events (see detailed discussion below). Consequently, for the case study presented herein, the presence of non-stationary methane outburst events did not lead to systematic biases in the EC-based long-term methane budget that go beyond the regular measurement uncertainty.

At tower 2, at times without event occurrence the EC-derived fluxes overestimated the wavelet reference by 1.2 % under high stationarity, and 5.8 % under medium stationarity. Similar offsets were observed at tower 1 (not shown). During events, the overestimation of fluxes under medium stationarity (11 %) approximately matched these biases, while under high stationarity, fluxes tended to be underestimated by 18 %. At tower 1, on the other hand, event fluxes under both stationarity categories were underestimated by 9 – 13 %. With the contributions of total fluxes per stationarity category ranging between

M G 5/29/2019 21:30
Deleted: . ... [1]

M G 5/29/2019 21:32
**Moved up [1]:** As a consequence, in the absence of process-based algorithms all gapfilling methods are dependent on the distribution of gaps to be filled, and therefore their performance is subject to a certain level of randomness.

M G 5/27/2019 21:30
**Deleted:** (fraction of QF7-8,

M G 6/11/2019 10:45
**Deleted:** Table 2

M G 6/11/2019 10:45
**Deleted:** Table 4

M G 5/29/2019 21:33
**Deleted:** With more gaps, first the database to train the algorithms would shrink, while second any systematic bias in gapfilling output would gain importance for the overall budget computation.

M G 5/29/2019 21:35
**Deleted:** fact that

M G 5/29/2019 21:35
**Deleted:** the

M G 5/29/2019 21:35
**Deleted:** is higher

0.7 – 2.8 % across towers, this minor tendency towards underestimating event fluxes did not influence the EC-computed flux budgets considerably.

During low stationarity conditions, all fluxes based on EC-processing will be sorted out, and will subsequently be replaced by gapfilling values, independent whether or not an event was contained in the specific half-hourly window. Therefore, the
correspondence between gapfilling results and wavelet reference fluxes was largely identical between 'events' and 'no events' cases at both towers. The influence of events under such circumstances is therefore restricted to the question whether or not event occurrences increase the fraction of detected low stationarity cases, which will be filtered out during quality screening and therefore create gaps. Data summarized in Table 5 show that, for our dataset from tower 2, the relative fraction of cases with low stationarity was ~31 % across half-hourly fluxes that were influenced by events, compared to only 4.7 %
for 'no events' cases. This observation indicates that, in general, more events hold the potential to cause more gaps in the flux time series, therefore with more events the gapfilling becomes more important.

Regarding the impact of events on the short-term variability of fluxes, the range of differences between wavelet- and EC-derived 30-minute flux rates is similar for 'events' and 'no events' cases (see Appendix D, Fig. A3); however, while during 'no events' cases a large number of values still shows good correspondence, those cases with substantial deviations from the
1:1 line dominate the method intercomparison for fluxes influenced by events. This is clearly indicated by the root mean square errors (Table A2), which under all stability categories are higher for the events cases. Under high to medium stationarity, the offsets produced by EC-processing appear to be random, therefore the number of events does not seem to introduce a systematic bias into the long-term budget. Still, Figure A3 demonstrates that, particularly for medium stationarity, the EC-derived flux rates influenced by events have a poor quality overall, with RMSE values >40 nmol m$^{-2}$ s$^{-1}$
found for both towers.

Our findings demonstrate that regular eddy-covariance flux processing yields highly reliable results under high stationarity conditions, while for medium to low stationarity, the half-hourly averaged flux rates by the wavelet method should be preferred instead when investigating methane emission dynamics at high temporal resolution. Particularly in the presence of events, individual EC flux rates are associated with a very high uncertainty, and should only be used for the computation of
long-term flux budgets. With events often occurring at timescales of only a few minutes, the wavelet flux processing holds the potential to provide new insights into the characteristics of these important elements of the methane cycle, since it facilitates flux computation down to timesteps of one minute without violating underlying theoretical assumptions. As demonstrated already for the decomposition of flux signals from spatially varying source areas (Metzger et al., 2013; Xu et al., 2017), wavelets provide a valuable tool for investigating the statistics of highly irregular emissions, how they can be
correlated with environmental conditions, and potentially be resolved by process-based algorithms for gapfilling and/or extrapolation purposes.

**M G 5/29/2019 21:36**
**Deleted:** If different environmental conditions promoted a higher event fraction, this effect may contribute to at least partly balancing the overestimation of fluxes while no events are present. Since fluxes under high and medium stationarity will not be flagged by the EC quality control, these offsets between 'events' and 'no events' fluxes directly influence the sums of the long-term methane budgets.

**M G 6/11/2019 10:45**
**Deleted:** Table 5

**M G 5/29/2019 21:37**
**Deleted:** , a difference by a factor >6

**M G 5/29/2019 21:37**
**Deleted:** How much this effect would influence the long-term carbon budget is in turn depending on the performance of the gapfilling routines, as discussed in Section 4.3.

**M G 5/29/2019 20:52**
**Deleted:** C

**M G 5/29/2019 21:38**
**Deleted:** Accordingly, it follows that an increasing fraction of events in a methane flux dataset will also reduce the level of correlation between wavelet reference and EC-fluxes.

**M G 5/29/2019 21:38**
**Deleted:** The low stationarity cases shown in Figure A3 do not contribute to this discussion, since for both 'no events' and 'events' cases, gapfilling values have been used, so both are subject to a high level of scatter.

**M G 5/27/2019 15:41**
**Deleted:** W

**M G 5/27/2019 15:41**
**Deleted:** therefore

**M G 5/27/2019 15:41**
**Deleted:** such

**M G 5/27/2019 15:42**
**Deleted:** bursts

## 5 Conclusions

Our study investigated the impact of short-term episodic emission outbursts, so-called event fluxes, on the overall data quality of methane fluxes observed by eddy-covariance towers over a wet tussock tundra ecosystem in Northeast Siberia. We evaluated the EC flux dataset against reference fluxes based on wavelet processing, which are not restricted to stationary flow conditions, and can resolve flux patterns down to timesteps of one minute. The wavelet analysis demonstrates that high methane emission events influenced 3 – 4 % of the flux observations during our study period, with integrated event emissions contributing 3 – 6 % to the net methane budget. EC flux data processing tended towards slightly underestimating the wavelet fluxes while events were present, but the net impact on long-term flux budgets is minor in relation to other uncertainties associated with eddy-covariance measurements. For the intercomparison of flux rates at 30-minute timesteps, however, our results demonstrate that the presence of events substantially increases the scatter between wavelet- and EC-derived fluxes, indicating that events introduce additional uncertainty into the EC-results.

A second focus of this study was placed on the evaluation of common gapfilling approaches for EC-derived methane fluxes. Our wavelet-derived fluxes provided an observation-based reference for the fraction of gaps in the EC time series that was created because measurements under low stationarity were filtered out by the data quality assessment protocol. None of the three gapfilling approaches tested herein could reproduce the range of values provided by the wavelet reference, but resulting biases in long-term flux budgets were still minor because of the comparatively small fraction of gaps that needed to be filled in our datasets. The performance of the gapfilling methods appeared to be dependent on the gap distribution, and the ratio of flux rates between the gaps and the remaining dataset. With a profound mechanistic understanding on processes and controls that govern the short-term variability in methane emissions still lacking, the quality of gapfilling products retains a certain level of randomness, therefore systematic biases even over longer timeframes cannot be ruled out particularly for datasets that contain a higher gap fraction as the ones used in our study.

Our findings demonstrate that wavelet analyses hold the potential to enhance our understanding in methane exchange processes between terrestrial ecosystems and the atmosphere. With excellent agreement between wavelet- and EC-derived fluxes demonstrated under ideal turbulence conditions, wavelet fluxes facilitate to quantify biases in EC-datasets linked to non-ideal conditions, e.g. medium to low stationarity of the flow. Moreover, the provision of observationally-based reference fluxes at times when the EC-method produces data gaps can support the development of novel process-based modeling algorithms that are representative for a wider range of environmental conditions, which can be employed e.g. for the improvement of gapfilling algorithms. Finally, the option to resolve fluxes down to temporal resolutions of one minute facilitates new insights into the intermittent nature of methane emissions, and its impact on the quality of methane flux observations.

**Acknowledgements**

The authors would like to thank Prof. Dr. Thomas Foken (University of Bayreuth) for his comments to an earlier version of the manuscript. This work was supported through funding by the European Commission (PAGE21 project, FP7-ENV-2011, Grant Agreement No. 282700; PerCCOM project, FP7-PEOPLE-2012-CIG, Grant Agreement No. PCIG12-GA-201-333796; INTAROS project, H2020-BG-09-2016, Grant Agreement No. 727890, Nunataryuk project, H2020-BG-11-2016/17, Grant Agreement No. 773421), the German Ministry of Education and Research (CarboPerm project, Grant No. 03G0836G; KoPf project, Grant No. 03F0764D), and the AXA Research Fund (PDOC_2012_W2 campaign, ARF fellowship M. Göckede). Furthermore the German Academic Exchange Service (DAAD) provided financial support for the travel expenses. The authors appreciate the contribution of staff members of the Northeast Scientific Station in Chersky for facilitating the field experiments.

**Software availability**

The software scripts that execute the wavelet-based flux data processing can be made available by the authors upon request.

**Author contributions**

MG was responsible for study conception and supervision. All authors contributed to development of the methodology and formal data analysis, with computation largely carried out by FK (eddy covariance) and CS (wavelets). All authors contributed to underlying fieldwork. MG wrote the initial manuscript, with contributions by FK. All authors contributed to reviewing the manuscript text, and editing the final manuscript version.

## Appendix A: Wavelet approach to calculate turbulent fluxes

The following description of the wavelet method is a slightly shortened version of the description provided by Schaller et al. (2017), which is the companion paper introducing the methodology that the presented study is based upon. It has been included here again to facilitate an easier overview on the procedure, without having to read other manuscripts. For more details, please refer to Schaller et al. (2017).

A continuous wavelet transform of a discrete time series $x(t)$ can be written as convolution of $x(t)$,

$$T(a,b) = \int_{-\infty}^{\infty} x(t) \cdot \psi_{a,b}^*(t) dt \qquad \text{(Eq. A1)}$$

where $T(a,b)$ is the wavelet coefficient and $\Psi_{a,b}(t)$ is referred to as wavelet function

$$\Psi_{a,b}(t) = \frac{1}{\sqrt{a}} \cdot \Psi\left(\frac{t-b}{a}\right). \qquad \text{(Eq. A2)}$$

The wavelet $\Psi$ requires a dilation parameter $a$, which controls the scale of the wavelet and thus the current frequency of interest, and a translation parameter $b$ that indicates the temporal position of the wavelet in the time series. For as complex-valued wavelet, the conjugate $\Psi_{a,b}^*(t)$ denoted by a star sign is used.

As mentioned in the main text above, this study used the complex-valued Morlet wavelet for quantification of flux rates, and the Mexican Hat wavelet for the exact localization of $CH_4$ emission events (see Schaller et al., 2017 for more details). The expression $T^2(a,b)$ across all times and scales provides the total energy of the time series. The average of the wavelet scalogram $|T^2(a,b)|$ is used to obtain the wavelet spectrum (Torrence and Compo, 1998)

$$E_x(j) = \frac{\delta t}{c_\delta} \cdot \frac{1}{N} \cdot \sum_{n=0}^{N-1} |T^2(a,b)| \qquad \text{(Eq. A3)}$$

over a given number $N$ of values in the time series, taking the time step $\delta t$ and a wavelet-specific reconstruction factor $C_\delta$ into account. From this it is now possible to obtain the global variance of the time series, $\sigma_x^2$, by integrating over all scales $j = 0$ to $J$

$$\sigma_x^2 = \frac{\delta t}{c_\delta} \cdot \frac{\delta j}{N} \cdot \sum_{n=0}^{N-1} \sum_{j=0}^{J} \frac{|T^2(a,b)|}{a(j)} \qquad \text{(Eq. A4)}$$

with $\delta_j$ referring to the spacing between discrete scales and $J$ being the maximum number of scales.

For two simultaneously recorded time series $x(t)$ and $y(t)$ the wavelet cross spectrum can now be obtained in analogy to Eq. (A3) as

$$E_{xy}(j) = \frac{\delta t}{c_\delta} \cdot \frac{1}{N} \cdot \sum_{n=0}^{N-1} [T_x(a,b) \cdot T_y^*(a,b)]. \qquad \text{(Eq. A5)}$$

where $T_y^*(a,b)$ denotes the complex conjugate of the wavelet transform of the second time series $y(t)$ (Hudgins et al., 1993). Summing up over all scales yields the covariance (Stull, 1988)

$$\overline{x'y'} = \frac{\delta t}{c_\delta} \cdot \frac{\delta j}{N} \cdot \sum_{n=0}^{N-1} \sum_{j=0}^{J} \frac{[T_x(a,b) \cdot T_y^*(a,b)]}{a(j)} \qquad \text{(Eq. A6)}$$

for the chosen averaging interval. If the chosen time series $x$ and $y$ are the vertical wind velocity $w$ and a corresponding gas concentration $c$, the flux $\overline{w'c'}$ can be calculated now using Equation (A6).

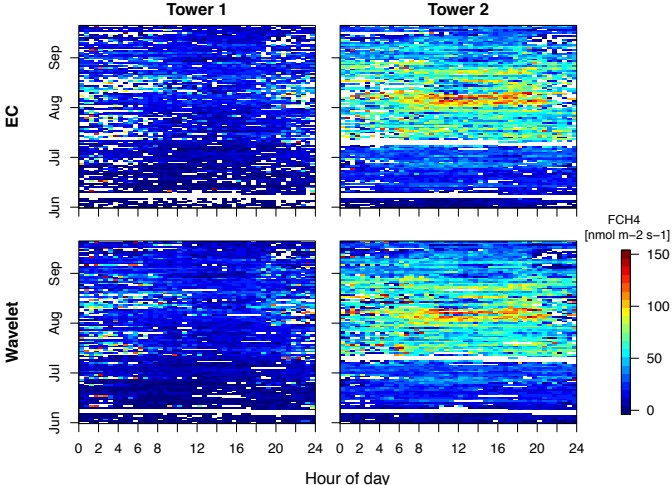

**Figure A1: Fingerprint plots showing the diurnal distribution of flux rates and gaps (white) for both towers and processing methods.**

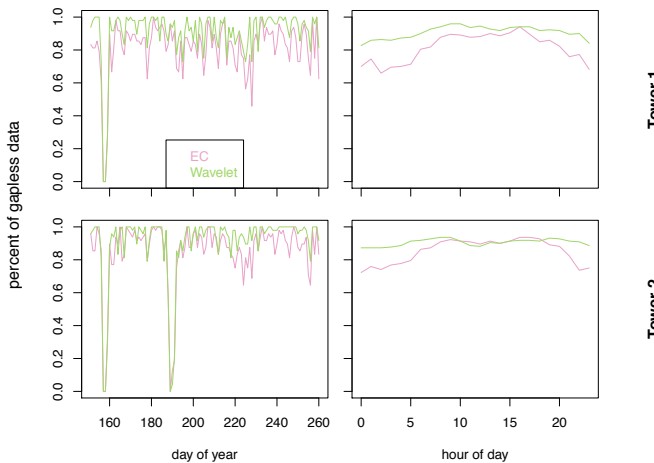

**Figure A2: Seasonal (left) and diurnal (right) distribution of data availability for wavelet- (green lines) and EC-derived (purple lines) methane fluxes.**

Table A1: Mean methane fluxes for the wavelet method and EC method, split into three stationarity categories. For the lowest stationarity, in addition to measured EC values, model results by the three different gapfilling approaches linear interpolation (LI), moving window (MW) and neuronal network (NN) are given. Absolute flux values are given in cells with white background, while grey shading indicates flux differences between EC- and wavelet processing, where numbers in brackets give the percentage deviation.

| | Flux calculation method | Peak [nmol m$^{-2}$ s$^{-1}$] | Updown/downup [nmol m$^{-2}$ s$^{-1}$] | Cluster [nmol m$^{-2}$ s$^{-1}$] | No event [nmol m$^{-2}$ s$^{-1}$] |
|---|---|---|---|---|---|
| **High stationarity** | Wavelet | 40.97 | 32.83 | 68.94 | 46.52 |
| | EC | 34.87 | 35.71 | 55.16 | 47.03 |
| | EC-Wavelet | -6.1 (-14%) | 2.88 (8.8%) | -13.78 (-20%) | 0.51 (1.1%) |
| **Medium stationarity** | Wavelet | 34.2 | 9.81 | 47.22 | 31.67 |
| | EC | 31.87 | 9.07 | 50.06 | 33.1 |
| | EC-Wavelet | -2.33 (-6.8%) | -0.74 (-7.5%) | 2.84 (6.0%) | 1.43 (4.5%) |
| **Low stationarity** | Wavelet | 42.42 | NA | 53.59 | 34.05 |
| | EC_measured | 31.5 | NA | 32.71 | 16.07 |
| | EC_gapfilled LI | 45.34 | NA | 65.17 | 36.8 |
| | EC_gapfilled MW | 50.28 | NA | 59.43 | 45.02 |
| | EC_gapfilled NN | 33.75 | NA | 45.22 | 35.72 |
| | EC_measured-Wavelet | -10.92 (-26%) | NA | -20.88 (-39%) | -17.98 (-53%) |
| | EC_gapfilled LI-Wavelet | 2.92 (6.9%) | NA | 11.58 (22%) | 2.75 (8.1%) |
| | EC_gapfilled MW-Wavelet | 7.86 (19%) | NA | 5.84 (11%) | 10.97 (32%) |
| | EC_gapfilled NN-Wavelet | -8.67 (-20%) | NA | -8.37 (-16%) | 1.67 (4.9%) |

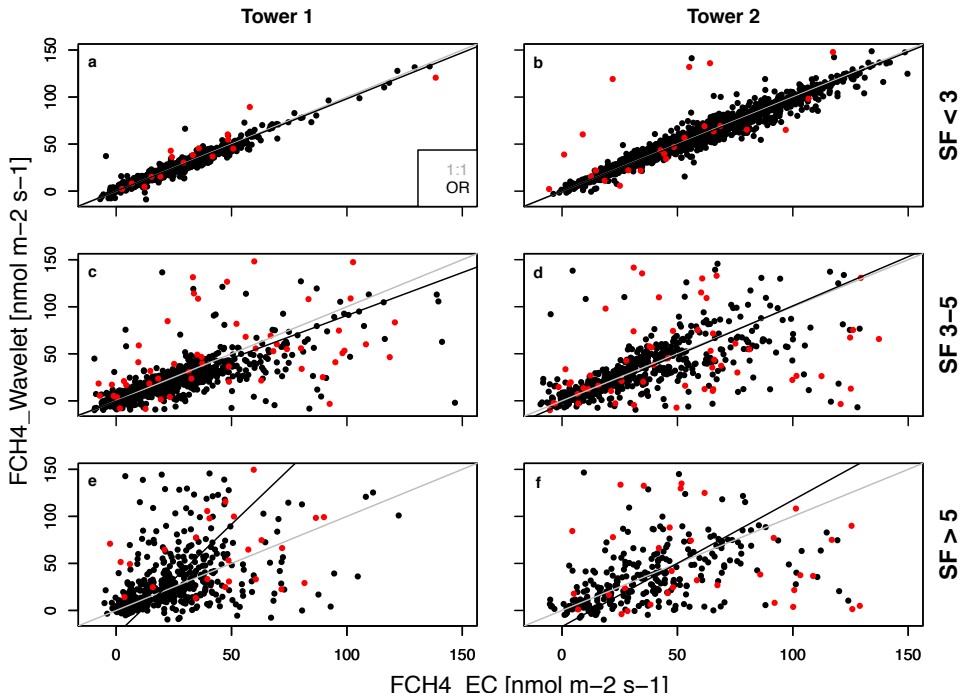

**Figure A3: Impact of events on the direct intercomparison of half-hourly flux rates between wavelet and eddy-covariance processing methods, sorted by tower and stationarity flag (SF) category. The displayed dataset includes gapfilled data, where linear interpolation was used to fill gaps for the EC method under low stationarity. Fluxes influenced by events are plotted in red, while 'no events' cases are black. The thin grey line gives the 1:1 line, the black line the fit of the orthogonal regression (OR) analysis.**

**Table A2: Deviations between 30-minute averaged fluxes based on wavelet- and EC-processing, expressed as the root mean square error [nmol m$^{-2}$ s$^{-1}$]. Time series include gapfilled values from linear interpolation for the EC time series.**

| | Tower 1 | | Tower 2 | |
|---|---|---|---|---|
| **Stationarity** | **No events** | **Events** | **No events** | **Events** |
| High (SF < 3) | 2.64 | 11.89 | 4.54 | 34.04 |
| Medium (SF 3 − 5) | 12.11 | 40.75 | 21.97 | 48.61 |
| Low (SF > 5) | 27.67 | 42.06 | 28.12 | 60.76 |

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
