# Peer review of "Quantifying the impact of emission outbursts and non-stationary flow on eddy covariance CH4 flux measurements using wavelet techniques"

_Biogeosciences, 2019_

## Referee Comment (RC1) · Gil Bohrer (Referee) · 2 Apr 2019

The manuscript studies a very relevant and much discussed problem - the challenge to methane flux measurements, given the high importance of "hot moments" and strong flux bursts. The analysis methods they use are robust and innovative, and the conclusions, particularly with regard to uncertainty and evaluation of different empirical approaches methane flux modeling (gap-filling) are very relevant and interesting.

I have a few minor comments: Introduction- P2. L5-15 You discuss ebullition as the major (and only one discussed) source of flux peaks. This may be the case, but I argue (with much support from observations by my group and others) that the spatial heterogeneity of methane fluxes can be interpreted as bursts and peaks when observed by the tower. For example, if there is a small patch that for whatever reason emits 2X or 5X more flux than the surrounding area, a small movement of the footprint to overlap more with that patch will read as a strong peak in emissions. I think most of what you define later as cluster events are driven by this spatial heterogeneity and not bubbling. That is very typical in wetlands, even within what would otherwise be considered a homogeneous land-cover type. Please discuss spatial heterogeneity as a source of flux spikes, not only temporal bursts.

P2.L30-35 Xu, Metzger and Desai 2017 AFM used wavelet flux calculation as the foundation for their "Upscaling tower-observed turbulent exchange at fine spatio-temporal resolution". Please check out what they did. It will make sense to reference that study here, but there are many parallel between their study and yours that should be acknowledged, some would fit later in the discussion.

Table 1 - the code in the table are meaningless outside the software package you used for flux processing. Can you provide equivalent physical ranges of something (standard deviation, thresholds to exceedance, % different before-after for stationarity ...) that will define these code and will make the table more meaningful? These codes define the analysis. Will be very important to define them using real-world (physical or statistical) conditions.
* * *

---

## Referee Comment (RC2) · Gil Bohrer (Referee) · 22 Apr 2019

I reread the paper. Got stuck on the last section of the methods where you say "Mexican hat wavelet" was used to determine bursts. That made me realize that I do not actually have a sense of what you are doing and how.

I think it'll be of great advantage for science, and the readers of Biogeosciences that you will post the code that you used to conduct the wavelet analyses, both for determining the flux, and for identifying bursts (I assume these are similar codes, with different setups, but may be wrong). A clear, well commented code, with a data example from your own study will go a long way in terms of applicability and citation number for this

paper.

Also (assuming your code is in R) consider making an R package and posting it in CRAN. But if that is too much of an effort, or if most of what you used is from other packages and all you did was set up and wrap up, please, at least, post the code with a working data example as appendix to this paper. Having a working example of how wavelet is used to determine fluxes from heterogeneous environments, and identify bursts would be great.

---

## Referee Comment (RC3) · Anonymous Referee #2 · 16 May 2019

Goeckede and others compare established eddy covariance and wavelet-based flux calculation techniques as well as gap filling techniques to measure methane flux - including episodic ebullition events - in an arctic ecosystem. The manuscript as written is acceptable for publication following minor improvements in my opinion.

Regarding the introduction, the case of ebullition extends beyond arctic ecosystem examples. Arctic ecosystems are of course important, but this approach can extend beyond them.

In section 2, please write scientific names in italics.

Does the filter on p 5 line 10 filter out many extreme values or many values close to

the thresholds? Just curious if methane ebulittion events may be excluded by this filter. (see also p. 17 L. 25).

Page 5 line 15 check 'NN, Dengel' reference.

I understand that the wavelet approach is described in detail elsewhere, but more detail in the present manuscript would help the reader grasp the basics of the approach without having to read other manuscripts to understand the present one.

From the results, do you suspect that atmospheric conditions may lead to ebullition events? In other words, does a Venturi effect occur with higher atmospheric wind speeds that results in pressure pumping? (see manuscripts by Bill Massman on this notion for soil and snow gas exchange).

The manuscript as a whole is cautious, insightful, and well-written but the Discussion section could use moderate restructuring so that it is a bit more succinct.

---

## Author Comment (AC1) · 4 Jun 2019

**Author response to interactive comment RC1 submitted by Gil Bohrer on Apr 02, 2019**

In the document below, the comments by Gil Bohrer have been copied from the original review and are shown in black font, while the author comments have been added in blue.

The manuscript studies a very relevant and much discussed problem - the challenge to methane flux measurements, given the high importance of "hot moments" and strong flux bursts. The analysis methods they use are robust and innovative, and the conclusions, particularly with regard to uncertainty and evaluation of different empirical approaches methane flux modeling (gap-filling) are very relevant and interesting.

I have a few minor comments: Introduction- P2. L5-15 You discuss ebullition as the major (and only one discussed) source of flux peaks. This may be the case, but I argue (with much support from observations by my group and others) that the spatial heterogeneity of methane fluxes can be interpreted as bursts and peaks when observed by the tower. For example, if there is a small patch that for whatever reason emits 2X or 5X more flux than the surrounding area, a small movement of the footprint to overlap more with that patch will read as a strong peak in emissions. I think most of what you define later as cluster events are driven by this spatial heterogeneity and not bubbling. That is very typical in wetlands, even within what would otherwise be considered a homogeneous landcover type. Please discuss spatial heterogeneity as a source of flux spikes, not only temporal bursts.

The authors agree that also spatial variability in $CH_4$ emission sources within the footprint of the flux sensor may lead to spikes in the flux time series. In the revised version of the manuscript, we will therefore include a new paragraph into the introduction section that acknowledges this influence of the spatial context.

In the companion paper by Schaller et al. (2019), the correlation of detected $CH_4$ outburst events with changes in environmental conditions was studied in detail. Here, no specific numbers are presented on the percentage of cases where the wind direction shifted substantially before and/or after an event. Still, several potential 'event triggers' are discussed that include a shift in wind direction, for example weather fronts. For most of these triggers, the dominating mechanism for a resulting change in flux rates is rather the change in atmospheric transport and turbulence conditions, as opposed to an associated shift in the field of view of the flux sensors.

Within the detailed analyses of 'peak' and 'up-down/down-up' events, we did not observe cases that could be attributed to an isolated shift in the footprint area, i.e. a shift in wind direction without changes in the turbulent flow field. The main reason to rule out the footprint effect is that all of these cases were observed simultaneously at the 2 towers that are located ~600m apart, and which clearly

feature different microscale patterns in CH$_4$ emission sources within the footprint area. For the category 'cluster events', none such detailed attribution studies were possible, so for these cases, a potential role of footprint shifts as an event trigger is possible. These cluster events should be studied in more detail in future, including extensions in the observational setup (see also Schaller et al., 2019). Summarizing, our site does not seem to be susceptible for footprint triggers because of the gradients between high and low CH$_4$ emission areas and their spatial structure. At other sites, however, the role of wind direction may be more pronounced.

P2.L30-35 Xu, Metzger and Desai 2017 AFM used wavelet flux calculation as the foundation for their "Upscaling tower-observed turbulent exchange at fine spatio-temporal resolution". Please check out what they did. It will make sense to reference that study here, but there are many parallel between their study and yours that should be acknowledged, some would fit later in the discussion.

The methodology applied by Xu et al. (2017) indeed shows overlaps with our own, but the objectives of both studies are quite different. Using the eddy flux processing package based on wavelets originally presented by Metzger et al. (2013), Xu et al. (2017) compute turbulent exchange fluxes at 1-minute resolution, which is very similar to employing the wavelet method presented by Schaller et al. (2017; 2019) as done for this study. Accordingly, we agree with Gil Bohrer that the paper by Xu et al. (2017) should be referenced herein, and we will include it in introduction and discussion.

However, Xu et al. (2017) used the higher temporal resolution in the flux time series to decipher the role of a varying field of view of the eddy tower on variability in the flux time series, this way avoiding aggregation errors linked to wind direction variability within the regular 30-minute flux processing timesteps. In our study, on the other hand, the primary objective was to use an alternative, wavelet-based processing approach to circumvent the need for stationary signals in flux processing, while as a secondary target we wanted to constrain highly intermittent CH$_4$ 'flux outbursts', and quantify potential biases linked to incorrect EC fluxes derived through regular flux processing.

Table 1 - the code in the table are meaningless outside the software package you used for flux processing. Can you provide equivalent physical ranges of something (standard deviation, thresholds to exceedance, % different before-after for stationarity ...) that will define these code and will make the table more meaningful? These codes define the analysis. Will be very important to define them using real-world (physical or statistical) conditions.

We agree that the overall quality flag rating in the form of numeric flags is of no importance in this context. We will therefore remove this column from Table 1, and will also delete all references to numeric overall quality flags from the main text. For the stationarity flags, we will add the numeric values of allowed percentage deviations that were defined by the quality control scheme devised by Foken et al. (2004; 2012) to Table 1.

**Cited references**

Foken, T., Göckede, M., Mauder, M., Mahrt, L., Amiro, B., and Munger, W.: Post-Field Data Quality Control, in: Handbook of Micrometeorology: A Guide for

Surface Flux Measurement and Analysis, edited by: Lee, X., Massman, W., and Law, B., Springer Netherlands, Dordrecht, 181-208, 2004.

Foken, T., Leuning, R., Oncley, S. P., Mauder, M., and Aubinet, M.: Corrections and data quality, in: Eddy Covariance - A practical guide to measurement and data analysis, edited by: Aubinet, M., Vesala, T., and Papale, D., Springer, Dordrecht; Heidelberg; London; New York, 85-131, 2012.

Metzger, S., Junkermann, W., Mauder, M., Butterbach-Bahl, K., Trancon y Widemann, B., Neidl, F., Schaefer, K., Wieneke, S., Zheng, X. H., Schmid, H. P., and Foken, T.: Spatially explicit regionalization of airborne flux measurements using environmental response functions, Biogeosciences, 10, 2193-2217, 2013.

Schaller, C., Göckede, M., and Foken, T.: Flux calculation of short turbulent events – comparison of three methods, Atmos. Meas. Tech., 10, 869-880, 2017.

Schaller, C., Kittler, F., Foken, T., and Göckede, M.: Characterisation of short-term extreme methane fluxes related to non-turbulent mixing above an Arctic permafrost ecosystem, Atmos. Chem. Phys., 19, 4041-4059, 2019.

Xu, K., Metzger, S., and Desai, A. R.: Upscaling tower-observed turbulent exchange at fine spatio-temporal resolution using environmental response functions, Agr. Forest Meteorol., 232, 10-22, 2017.

---

## Author Comment (AC2) · 4 Jun 2019

**Author response to interactive comment RC2 submitted by Gil Bohrer on Apr 22, 2019**

In the document below, the comments by Gil Bohrer have been copied from the original review and are shown in black font, while the author comments have been added in blue.

I reread the paper. Got stuck on the last section of the methods where you say "Mexican hat wavelet" was used to determine bursts. That made me realize that I do not actually have a sense of what you are doing and how.

I think it'll be of great advantage for science, and the readers of Biogeosciences that you will post the code that you used to conduct the wavelet analyses, both for determining the flux, and for identifying bursts (I assume these are similar codes, with different setups, but may be wrong). A clear, well commented code, with a data example from your own study will go a long way in terms of applicability and citation number for this paper.

Also (assuming your code is in R) consider making an R package and posting it in CRAN. But if that is too much of an effort, or if most of what you used is from other packages and all you did was set up and wrap up, please, at least, post the code with a working data example as appendix to this paper. Having a working example of how wavelet is used to determine fluxes from heterogeneous environments, and identify bursts would be great.

It is our goal to make the wavelet software package, which has been programmed in R, publicly available in the near future. This may be either in form of a stand-along R-package, or a code package in GitHub (or both). This goal, however, cannot be completed within the coming months, instead a publication of a 'cleaned up', self-explanatory code with all the necessary documentation is foreseen for fall 2019.

At the time of writing, a code version with reduced documentation is already available, but we do not want to provide this as an open-access package due to the above-mentioned plans to develop a better-documented version soon. The current version, however, is already sufficiently commented to allow the interested user to reproduce the results that have been presented in the manuscript, as well as in the companion papers by Schaller et al. (2017; 2019). The availability of this code by the authors upon request has been added as a note to the end of this manuscript.

**References**

Schaller, C., Göckede, M., and Foken, T.: Flux calculation of short turbulent events – comparison of three methods, Atmos. Meas. Tech., 10, 869-880, 2017.

Schaller, C., Kittler, F., Foken, T., and Göckede, M.: Characterisation of short-term extreme methane fluxes related to non-turbulent mixing above an Arctic permafrost ecosystem, Atmos. Chem. Phys., 19, 4041-4059, 2019.

---

## Author Comment (AC3) · 4 Jun 2019

**Author response to interactive comment RC3 submitted by an anonymous referee on May 16, 2019**

In the document below, the comments by the anonymous referee have been copied from the original review and are shown in black font, while the author comments have been added in blue.

Goeckede and others compare established eddy covariance and wavelet-based flux calculation techniques as well as gap filling techniques to measure methane flux - including episodic ebullition events - in an arctic ecosystem. The manuscript as written is acceptable for publication following minor improvements in my opinion.

Regarding the introduction, the case of ebullition extends beyond arctic ecosystem examples. Arctic ecosystems are of course important, but this approach can extend beyond them.

We will delete large parts of the first paragraph of the introduction to remove references to the Arctic in this general section of the manuscript. The remaining sentences will be merged subsequently with the first part of the second paragraph, this way creating a new first paragraph that generally focuses on eddy-covariance quality control issues, with a specific focus on methane.

In section 2, please write scientific names in italics.

This will be changed.

Does the filter on p 5 line 10 filter out many extreme values or many values close to the thresholds? Just curious if methane ebullition events may be excluded by this filter. (see also p. 17 L. 25).

Using the dataset from Tower 2 as an example, the range filter excluded 235 half-hourly data points in total. Of those, ~30 % were extreme negative outliers (fluxes < -50 nmol m$^{-2}$ s$^{-1}$, while the majority were moderate negative fluxes (47 % in the range -10 to -50 nmol m$^{-2}$ s$^{-1}$). Just ~14 % were strong positive outliers (150 – 250 nmol m$^{-2}$ s$^{-1}$), and the remaining ~8 % were extreme positive outliers (>250 nmol m$^{-2}$ s$^{-1}$). We acknowledge that the used threshold of 150 nmol m$^{-2}$ s$^{-1}$ is somewhat subjective, but an extension of that cutoff towards higher values would have had minor impact on the presented analysis.

Page 5 line 15 check 'NN, Dengel' reference.

We checked the record with Biogeosciences, but could not find any error in the previously used version ..??

I understand that the wavelet approach is described in detail elsewhere, but more detail in the present manuscript would help the reader grasp the basics of the approach without having to read other manuscripts to understand the present one.

We will add a new appendix A to the manuscript that describes the wavelet approach to calculate turbulent fluxes. The new material will be a shortened version of the original methods description as presented in the companion manuscript by Schaller et al. (2017).

From the results, do you suspect that atmospheric conditions may lead to ebullition events? In other words, does a Venturi effect occur with higher atmospheric wind speeds that results in pressure pumping? (see manuscripts by Bill Massman on this notion for soil and snow gas exchange).

Indeed, the pressure effect associated with changes in atmospheric transport and turbulence conditions, as described e.g. by Massman (2006) and Massman and Frank (2006), may play an important role for the occurrence of methane emission outbursts as analyzed in our study. For example, in the companion manuscript by Schaller et al. (2019) a case example of an emission event triggered by a passing weather front is described where the high methane releases are most likely caused by pressure pumping.

The manuscript as a whole is cautious, insightful, and well-written but the Discussion section could use moderate restructuring so that it is a bit more succinct.

We will revise the Discussion section, and target a new version that will be about 20% shorter than the current one.

**References**

Massman, W. J.: Advective transport of CO2 in permeable media induced by atmospheric pressure fluctuations: 1. An analytical model, J. Geophys. Res.-Biogeo., 111, 2006.

Massman, W. J., and Frank, J. M.: Advective transport of CO2 in permeable media induced by atmospheric pressure fluctuations: 2. Observational evidence under snowpacks, J. Geophys. Res.-Biogeo., 111, 2006.

Schaller, C., Göckede, M., and Foken, T.: Flux calculation of short turbulent events – comparison of three methods, Atmos. Meas. Tech., 10, 869-880, 2017.

Schaller, C., Kittler, F., Foken, T., and Göckede, M.: Characterisation of short-term extreme methane fluxes related to non-turbulent mixing above an Arctic permafrost ecosystem, Atmos. Chem. Phys., 19, 4041-4059, 2019.